# Separating Mesoscale and Submesoscale Flows from Clustered Drifter Trajectories

**Sarah Oscroft [1], Adam M. Sykulski [2],\* and Jeffrey J. Early [3]**

1    STOR-i Centre for Doctoral Training, Lancaster University, Lancaster LA1 4YW, UK;
     s.oscroft1@lancaster.ac.uk
2    Department of Mathematics and Statistics, Lancaster University, Lancaster LA1 4YW, UK
3    NorthWest Research Associates, Redmond, WA 98052, USA; jearly@nwra.com
\*    Correspondence: a.sykulski@lancaster.ac.uk

**Abstract:** Drifters deployed in close proximity collectively provide a unique observational data set with which to separate mesoscale and submesoscale flows. In this paper we provide a principled approach for doing so by fitting observed velocities to a local Taylor expansion of the velocity flow field. We demonstrate how to estimate mesoscale and submesoscale quantities that evolve slowly over time, as well as their associated statistical uncertainty. We show that in practice the mesoscale component of our model can explain much first and second-moment variability in drifter velocities, especially at low frequencies. This results in much lower and more meaningful measures of submesoscale diffusivity, which would otherwise be contaminated by unresolved mesoscale flow. We quantify these effects theoretically via computing Lagrangian frequency spectra, and demonstrate the usefulness of our methodology through simulations as well as with real observations from the LatMix deployment of drifters. The outcome of this method is a full Lagrangian decomposition of each drifter trajectory into three components that represent the background, mesoscale, and submesoscale flow.

**Keywords:** drifters; mesoscale; submesoscale; diffusivity; strain; vorticity; divergence; Lagrangian; frequency spectra; bootstrap; uncertainty quantification; splines

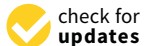



## 1. Introduction

Recent field experiments targeting submesoscale motions (100 m–10 km) include the deployment of dozens to hundreds of GPS tracked surface drifters in close proximity, e.g., 'LatMix' [1], 'GLAD' [2], 'LASER' [3] and 'CALYPSO' [4]. These deployments are designed to sample a narrow spatiotemporal window, but with high enough data density to resolve submesoscale motions. However, even when submesoscale motions are resolved, separating those motions from the larger, often more energetic mesoscale motions remains a significant challenge.

One approach to disentangling the submesoscales from the mesoscales with high resolution drifter data is to use the results from turbulence theory. For example, Ref. [2] showed results using two-particle statistics consistent with local dispersion at submesoscales. Ref. [5] found ambiguous results until inertial oscillations were filtered from the trajectories. This suggests, not surprisingly, that realistic flow fields contain a combination of flow features that can be linearly separated in some contexts. In a detailed modelling study, Ref. [6] showed that, even with some filtering, these Lagrangian statistics are far more sensitive than similar Eulerian measures, and called into question the interpretation of previous studies that use variations of two-particle statistics.

An alternative approach is to parameterise the energetic mesoscale flow features from the Lagrangian trajectories, in order to disentangle them from the unparameterised, possibly submesoscale, flows. The notion of accounting for, or parameterising, the mesoscale strain in order to measure the submesoscale diffusivity, appears to originate with tracer release experiments [7,8], and is based on ideas introduced in [9]. The basic idea is that one

axis of the tracer grows exponentially with a rate proportional to the strain rate, $\sigma$, while the other axis reaches a steady state balanced by the compressing effect of $\sigma$ and the elongating effect of diffusivity, $\kappa$. In the dye experiments, the mesoscale strain rate is determined by measuring the rate of elongation of the patch, which is then used to deduce the diffusivity. The key idea to this approach is that the mesoscale strain rate is parameterised, in order to separate its effect from the submesoscale motions.

This manuscript extends the idea of parameterising mesoscale features, in order to disentangle submesoscale flow, to a more principled and robust framework appropriate for Lagrangian particles. Our work is complementary to, but distinct from, the recent works of [3,10] who developed a method for projecting clustered drifter trajectories to reconstruct local Eulerian velocity fields using Gaussian Process regression. The goal of our work is to disentangle the trajectories in a Lagrangian sense, and explicitly separate each drifter trajectory into background, mesoscale and submesoscale components—where each decomposed drifter can then be analysed further within the Lagrangian framework. A key benefit is that our Lagrangian separation allows for the explicit estimation of submesoscale diffusivity as we shall show.

The structure of this paper is as follows. In Section 2, we first introduce a conceptual Lagrangian flow model, and then show how this can be parameterised using a local Taylor expansion. Then in Section 3 we show how these parameters can be estimated from clustered drifter deployments. We pay particular focus to building a hierarchy of models, where each layer in the hierarchy adds extra parameters (e.g. strain/vorticity/divergence) that represent additional flow features. We provide novel methodology for selecting between hierarchies based on the evidence from the data. In Section 4, we go further and incorporate nonstationary flow features, by allowing mesoscale parameters to slowly evolve over time. We provide methodology for estimating this evolution using splines, and then we provide techniques for quantifying the uncertainty of all parameter estimates using the bootstrap. We detail how this quantification of uncertainty provides the ideal mechanism from which to select the key parameter of the temporal window length. Throughout Sections 3 and 4 we perform detailed simulation analyses to provide further insight and motivation. Then in Section 5 we test and perform our novel methodologies on data collected from drifters in the LatMix deployment, which reveals new insights and discovers previously hidden mesoscale and submesoscale structures. Discussion and conclusions can be found in Section 6. We also perform a sensitivity analysis against the number of drifters, as well as the configuration of the initial deployment, in Appendix A. Code to replicate all results and figures in this paper is available at https://github.com/JeffreyEarly/GLOceanKit.

Overall, the principle contribution of this paper is a general framework for analysing Lagrangian data from clustered drifter deployments. Specifically, this methodology provides a tool to detect for the presence of various mesoscale flow features and separate those features from the submesoscale flow—while allowing such features to evolve over time—together with providing quantified statistical uncertainty of output.

## 2. Modelling Framework

The primary conceptual model used throughout this manuscript is that the total velocity of a Lagrangian particle $\mathbf{u}^{\text{total}}$ can be decomposed into three components,

$$\mathbf{u}^{\text{total}} = \mathbf{u}^{\text{bg}} + \mathbf{u}^{\text{meso}} + \mathbf{u}^{\text{sm}}, \tag{1}$$

where $\mathbf{u}^{\text{bg}}$ is a large scale background flow, $\mathbf{u}^{\text{meso}}$ is the mesoscale flow ($>$10 km, $>$10 days) and $\mathbf{u}^{\text{sm}}$ is the submesoscale flow (100 m–10 km, 1 h–10 days). The background flow is assumed to be spatially homogeneous in some local region around the drifters, and thus includes motions such as inertial oscillations and large scale currents. The terminology used here is appropriate for a range of oceanographic contexts, but arguably the separation into mesoscale and submesoscale are more precisely related to non-local and local dynamics, respectively. We thus use the term mesoscale to describe structures that behave non-locally

across the drifters, and are therefore the smoothly varying fluid structures that will be parameterised, such as the constant strain rate used in the tracer release experiments [8]. The submesoscale currents are simply the residual motion, not captured by the background or mesoscale flow. If any statistically significant submesoscale signal remains, its energy spectrum will likely be shallower than the mesoscale portion and therefore be consistent with local dynamics. In practice, the scales captured by these three types of motion will vary depending on the deployment details and the limitations of the data, as much as the actual physical processes themselves, as we shall show. The proposed methodology therefore ultimately remains agnostic to the scales and physical processes governing the motions, but instead focuses on the statistical significance of the model.

Surface drifter motion is constrained to a fixed depth near the ocean surface, where the two-dimensional positions are measured in geographic coordinates longitude and latitude. For the work here it is necessary to use map coordinates $\{x(t), y(t)\}$ with a projection that locally preserves area and shape. Following [11] we use the transverse Mercator projection with central meridian placed between the minimum and maximum longitude of the drifter experiment and add a false northing and easting to shift the origin to the southwest corner. The total velocity $\mathbf{u}^{\text{total}}$ of a drifter is then two-dimensional and assumed to represent the velocity at the depth of the drifter drogue. The work here will also be generally applicable to clustered deployments of RAFOS floats with minor modification, but we will use the terminology of drifters throughout the manuscript.

### 2.1. Local Taylor Expansion

One of the simplest models for separating flow components is to perform a local Taylor expansion of the velocity field. Suppose we have observations from $K$ clustered drifters at time $t$, where the position of drifter $k$ ($1 \leq k \leq K$) in $x$ and $y$ orthogonal directions is given by $\{x_k(t), y_k(t)\}$, measured in metres, and the corresponding velocity is given by $\frac{d}{dt}\{x_k(t), y_k(t)\}$, measured in metres per second. We then take a Taylor series expansion of the velocity field evaluated at the position of drifter $k$, such that we model its velocity as

$$\underbrace{\frac{d}{dt}\begin{bmatrix} x_k(t) \\ y_k(t) \end{bmatrix}}_{\mathbf{u}^{\text{total}}} = \underbrace{\begin{bmatrix} u^{\text{bg}}(t) \\ v^{\text{bg}}(t) \end{bmatrix}}_{\mathbf{u}^{\text{bg}}} + \underbrace{\begin{bmatrix} u_0 + u_1 t \\ v_0 + v_1 t \end{bmatrix} + \frac{1}{2}\begin{bmatrix} \sigma_n + \delta & \sigma_s - \zeta \\ \sigma_s + \zeta & \delta - \sigma_n \end{bmatrix}\begin{bmatrix} x_k(t) - x_0 \\ y_k(t) - y_0 \end{bmatrix}}_{\mathbf{u}^{\text{meso}}} + \underbrace{\begin{bmatrix} u_k^{\text{sm}}(t) \\ v_k^{\text{sm}}(t) \end{bmatrix}}_{\mathbf{u}^{\text{sm}}}, \quad (2)$$

where

- $\{x_k(t), y_k(t)\}$ are observations from drifter $k$ at time $t$;
- $\{u^{\text{bg}}(t), v^{\text{bg}}(t)\}$ is the spatially homogeneous time-varying background flow;
- $\{u_0, v_0, u_1, v_1, \sigma_n, \sigma_s, \zeta, \delta\}$ are the model parameters for the mesoscale flow;
- $\{x_0, y_0\}$ is the expansion location and has no consequence to the model, other than re-defining $\{u_0, v_0\}$;
- $\{u_k^{\text{sm}}(t)\ v_k^{\text{sm}}(t)\}$ are the residual 'submesoscale' velocities for each drifter, assumed to be zero-mean in time, but also zero-mean in space across drifters.

The mesoscale parameters are simply re-definitions of the standard spatial gradients: the divergence is $\delta = u_x + v_y$, the vorticity is $\zeta = v_x - u_y$, the normal strain rate is $\sigma_n = u_x - v_y$, and the shear strain rate is $\sigma_s = v_x + u_y$. The normal and shear strain rates can be combined to a scalar value for the strain rate $\sigma = \sqrt{\sigma_n^2 + \sigma_s^2}$ and rotation angle $\theta = \arctan\left[\sigma_s / \sigma_n\right]/2$, where $\sigma_n = \sigma \cos(2\theta)$, $\sigma_s = \sigma \sin(2\theta)$.

Equation (2) therefore separates background, mesoscale, and submesoscale features in the data, following the conceptual model of Equation (1). For the moment, the eight mesoscale parameters are assumed to be sufficiently slowly varying that we can treat them as constant over some time window, although we will relax this restriction later. In practice, the mesoscale component of the model will capture any coherent feature that has constant spatial gradient across the cluster of drifters, whether that is a large scale more permanent feature like a Western boundary current or a transient mesoscale eddy—or nothing at all. The spatially homogeneous time-varying 'background' flow will capture inertial and tidal

oscillations, but may also erroneously include parts of a time or spatially varying mesoscale flow. Finally, the residual 'submesoscale' velocity will include any velocity contributions not captured by the other components.

The model of Equation (2) was applied to drifter observations in [12] to obtain estimates of the spatial gradient parameters, but with two key differences from the approach taken here. First, the spatial gradients were allowed to vary at each observational time point, without any constraints on the rate of fluctuation. Second, the expansion point $\{x_0, y_0\}$ was chosen to be the time-varying centre-of-mass of the cluster of drifters. The consequence of this choice is quite significant and is worth considering in more detail. Defining the centre-of-mass (or first moment) as $m_x(t) \equiv \frac{1}{K} \sum_{k=1}^{K} x_k(t)$ and $m_y(t) \equiv \frac{1}{K} \sum_{k=1}^{K} y_k(t)$, it follows from Equation (2) that the centre-of-mass velocity includes contributions from both the homogeneous background as well as the spatial gradients such that

$$\frac{d}{dt} \begin{bmatrix} m_x(t) \\ m_y(t) \end{bmatrix} = \begin{bmatrix} u^{\mathrm{bg}}(t) \\ v^{\mathrm{bg}}(t) \end{bmatrix} + \begin{bmatrix} u_0 + u_1 t \\ v_0 + v_1 t \end{bmatrix} + \frac{1}{2} \begin{bmatrix} \sigma_n + \delta & \sigma_s - \zeta \\ \sigma_s + \zeta & \delta - \sigma_n \end{bmatrix} \begin{bmatrix} m_x(t) - x_0 \\ m_y(t) - y_0 \end{bmatrix}, \tag{3}$$

where no submesoscale is assumed to be present as we have defined $\frac{1}{K} \sum_{k=1}^{K} u_k^{\mathrm{sm}}(t) = 0$. That the mesoscale spatial gradients have a (potentially) significant impact on the velocity of the centre-of-mass is evident in the top row of simulated drifter trajectories shown in Figure 1, where the entire cluster of drifters is advected by the linear flow. Now if the expansion point is taken to be the centre-of-mass, $\{x_0(t), y_0(t)\} = \{m_x(t), m_y(t)\}$, then Equation (3) reduces the background velocity to the sample mean velocity, such that $u^{\mathrm{bg}}(t) \approx \frac{d}{dt} m_x(t)$. As a result, after subtracting Equation (3) from (2), the velocities of the individual particles in the centre-of-mass frame,

$$\frac{d}{dt} \begin{bmatrix} x_k(t) - m_x(t) \\ y_k(t) - m_y(t) \end{bmatrix} = \frac{1}{2} \begin{bmatrix} \sigma_n + \delta & \sigma_s - \zeta \\ \sigma_s + \zeta & \delta - \sigma_n \end{bmatrix} \begin{bmatrix} x_k(t) - m_x(t) \\ y_k(t) - m_y(t) \end{bmatrix} + \begin{bmatrix} u_k^{\mathrm{sm}}(t) \\ v_k^{\mathrm{sm}}(t) \end{bmatrix}, \tag{4}$$

only depend on the spatial gradients and submesoscale flow. In some sense, the difference between Equations (2) and (4) is quite remarkable: simply by changing to centre-of-mass coordinates, the potentially complicated form of the background flow, $\{u^{\mathrm{bg}}, v^{\mathrm{bg}}\}$, is eliminated, along with all the velocity variance associated with mesoscale advection of the centre-of-mass from Equation (3). With this choice of reference frame, the spatial gradients in the model now only characterise the spreading of particles, i.e., the second moment, as shown in the second row of Figure 1, along with any spreading caused by the submesoscale process.

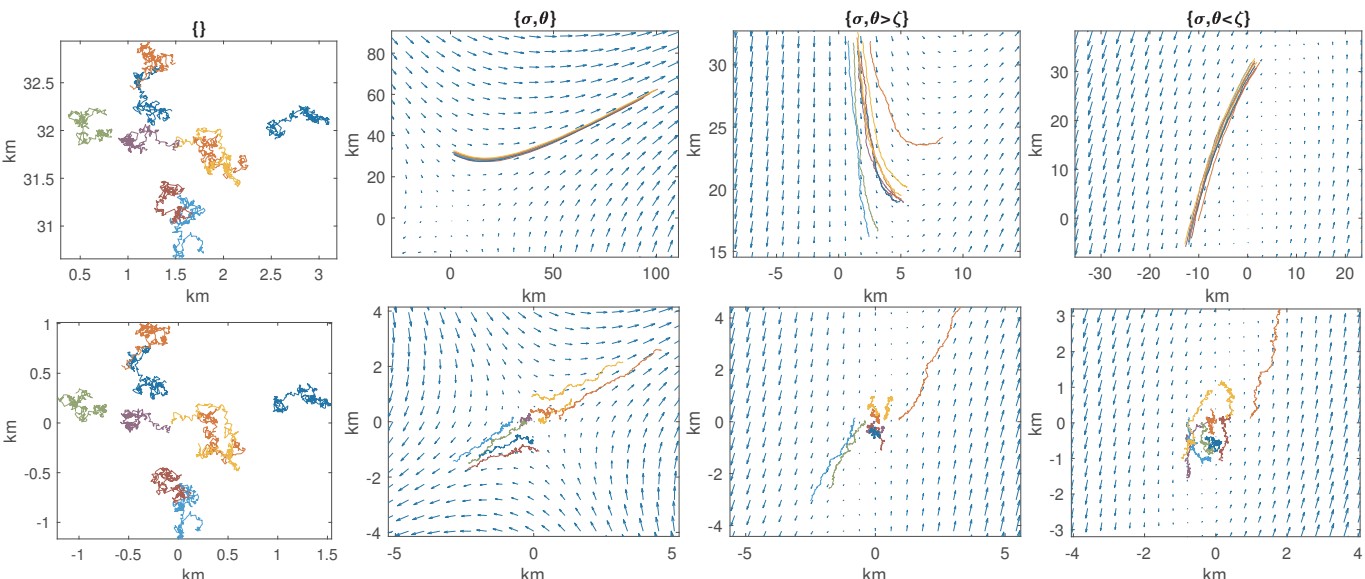

**Figure 1.** Simulation of nine drifters from Equation (2) over 6.25 days, with starting positions, number of drifters, and experiment length taken to match LatMix Site 1. In each panel the submesoscale velocities $\{u_k^{\text{sm}}(t), v_k^{\text{sm}}(t)\}$ follow a Wiener increment process with diffusivity equal to 0.1 m$^2$/s. The top row shows drifter positions, and the bottom row shows positions with respect to centre-of-mass at each time step. From left to right we include the following model components. Left: diffusivity only. Centre left: strain and diffusivity. Centre right: strain, vorticity, and diffusivity (strain dominated). Right: strain, vorticity, and diffusivity (vorticity dominated). In each plot where a parameter is present, it has been set as $\sigma = 7 \times 10^{-6}$/s, $\theta = 30°$, $\zeta = 6 \times 10^{-6}$/s (centre right), and $\zeta = 8 \times 10^{-6}$/s (right). We have set $u_0 = v_0 = u_1 = v_1 = u^{\text{bg}} = v^{\text{bg}} = 0$. The trajectories are simulated using the Euler–Maruyama scheme [13] and we include quivers in all plots representing the underlying velocity field.

### 2.2. Diffusivity

A key measure with which we evaluate our techniques is to measure the diffusivity of observed and modelled velocities. We define the submesoscale diffusivity for each drifter $k$ as in Equation (21) of [14], such that

$$\kappa_{k,x}^{\text{sm}}(t) = \frac{1}{2}\frac{d}{dt}x_k^{\text{sm}}(t)^2 = \int_0^t u_k^{\text{sm}}(t)u_k^{\text{sm}}(\tau)d\tau, \tag{5a}$$

$$\kappa_{k,y}^{\text{sm}}(t) = \frac{1}{2}\frac{d}{dt}y_k^{\text{sm}}(t)^2 = \int_0^t v_k^{\text{sm}}(t)v_k^{\text{sm}}(\tau)d\tau, \tag{5b}$$

where $x_k^{\text{sm}}(t)$ is calculated from residual velocities, $u_k^{\text{sm}}(t)$, such that $x_k^{\text{sm}}(t) = \int_0^t u_k^{\text{sm}}(t)dt$, and similarly for $y_k^{\text{sm}}(t)$. As in Equation (10) of [14], a joint diffusivity measure across all drifters could be defined by averaging the positions/velocities before applying the derivatives/integrals in Equations (5a) and (5b); however, we initially choose to calculate diffusivity separately for each drifter $k$ to reflect the fact that drifters are spatially spread in a clustered deployment, and hence their diffusivity values may depend on spatial scale within a spatially inhomogeneous flow field.

In general, it is also useful to consider the isotropic diffusivity as this is rotationally invariant, and as such, does not depend on our choice of coordinate system. The isotropic submesoscale diffusivity for drifter $k$ is defined as

$$\kappa_{k,z}^{\text{sm}}(t) = \frac{1}{4}\frac{d}{dt}z_k^{\text{sm}}(t)^2 = \frac{1}{2}\int_0^t w_k^{\text{sm}}(t)w_k^{\text{sm}}(\tau)d\tau, \tag{6}$$

where $z_k^{\text{sm}}(t) = x_k^{\text{sm}}(t) + iy_k^{\text{sm}}(t)$, $w_k^{\text{sm}}(t) = u_k^{\text{sm}}(t) + iv_k^{\text{sm}}(t)$, and $i \equiv \sqrt{-1}$. The isotropic diffusivity is the average of $\kappa_{k,x}^{\text{sm}}(t)$ and $\kappa_{k,y}^{\text{sm}}(t)$ such that $\kappa_{k,z}^{\text{sm}}(t) = \frac{1}{2}\{\kappa_{k,x}^{\text{sm}}(t) + \kappa_{k,y}^{\text{sm}}(t)\}$.

The diffusivity is also related to the power spectral density of complex velocity $w_k(t)$ where

$$S(\omega) \equiv \frac{1}{T} \left| \int_0^T w_k(t) e^{-\mathrm{i}\omega t} dt \right|^2. \tag{7}$$

$S(\omega)$ is known as the Lagrangian frequency spectrum and is related the isotropic diffusivity in Equation (6) with

$$\kappa_{k,z}(T) = \frac{1}{4} S(0), \tag{8}$$

as shown in [15]. Formally, diffusivity requires the process to be stationary and is defined in the limit as $T \to \infty$, but in practice we are always limited to finite observation times. The total variance of a complex particle velocity is conserved with the Lagrangian frequency spectrum, $\frac{1}{T} \int w_k(t)^2 dt = \int S(\omega) d\omega$, and in this sense it will be useful to think of how the model components in Equation (2) each describe the distribution of variance in the frequency spectrum.

Equations (5)–(7) are theoretical constructs as they require submesoscale velocities to be observed continuously in time. In practice, drifter observations are only observed at discrete time points. In Section 3, we will discuss how to estimate submesoscale diffusivity from clustered drifter data using our modelling and estimation approach.

We note that diffusivities could also be calculated directly from raw velocities $\{\frac{d}{dt}x_k(t),$ $\frac{d}{dt}y_k(t)\}$, or from centre-of-mass velocities that have only had the background removed and still contain mesoscale flow contribution (as in Equation (4)), and such values of diffusivity will in general be much larger than the submesoscale diffusivities. This highlights the scale-dependent nature of diffusivity, as well as the challenges in comparing different measurements of diffusivity.

### 2.3. Model Solutions

The mesoscale component of Equation (2) is a linear ordinary differential equation with tractable analytical solutions, e.g., [16,17]. However, the submesoscale component of Equation (2) is assumed unknown, and may represent a range of different phenomena. Thus, for our simulation analyses that follow in this paper we generate the submesoscale process stochastically using trajectory paths defined by

$$\frac{d}{dt} \begin{bmatrix} x_k(t) \\ y_k(t) \end{bmatrix} = \begin{bmatrix} u_0 \\ v_0 \end{bmatrix} + \frac{1}{2} \begin{bmatrix} \sigma_n + \delta & \sigma_s - \zeta \\ \sigma_s + \zeta & \delta - \sigma_n \end{bmatrix} \begin{bmatrix} x_k(t) - x_0 \\ y_k(t) - y_0 \end{bmatrix} + \sqrt{2\kappa}\mathbf{dW}_t, \tag{9}$$

where the function $\mathbf{dW}$ represents an increment of a two-dimensional Wiener process (a random walk in the discrete-time limit) that forms the submesoscale component. The Lagrangian frequency spectrum of the submesoscale process is therefore simply that of a white noise process:

$$S(\omega) = 4\kappa. \tag{10}$$

The frequency spectrum of internal waves (perhaps the best known submesoscale process) will have either more or less contribution to the total variance, depending on the frequency. We thus consider a white noise velocity process to be a reasonably agnostic choice. Notably absent from Equation (9) is the spatially homogeneous background flow. In practice this contains a significant amount of power from inertial and tidal oscillations, but does not significantly impact the estimation of mesoscale quantities as we shall show. The particle trajectories shown in Figure 1 are sampled from Equation (9), where each column contains different choices for the mesoscale parameters, but the submesoscale diffusivity $\kappa$ is held constant (the first column has no mesoscale and hence the particles follow a random walk).

In the absence of the stochastic submesoscale white noise process, the Lagrangian trajectories from Equation (9) are purely deterministic and thus their Lagrangian frequency spectra can be computed exactly, as we shall now show. For the following analytical solu-

tions we set $\delta = 0$, but make no such assumption in the estimation procedure that follows. To integrate Equation (9) with $\kappa = 0$, note that simply re-positioning a particle's initial location can be used to redefine $\{u_0, v_0\}$. Specifically, if the initial position of the particle is given by $\{x(0), y(0)\}$ with nonzero $\{u_0, v_0\}$, the $\{u_0, v_0\}$ can be set to zero, so long as the initial position is set to $\{x(0) - x_u, y(0) - y_u\}$ where

$$\begin{bmatrix} x_u \\ y_u \end{bmatrix} = \frac{2}{s^2} \begin{bmatrix} \sigma_n & \sigma_s - \zeta \\ \sigma_s + \zeta & -\sigma_n \end{bmatrix} \begin{bmatrix} u_0 \\ v_0 \end{bmatrix}, \tag{11}$$

and the Okubo–Weiss parameter is defined by $s^2 \equiv \sigma^2 - \zeta^2$. Thus, without loss of generality, we can simply take $\{u_0, v_0\}$ and the expansion point to be zero. The complex path $z(t) = x(t) + iy(t)$ with initial position given by $\{x(0), y(0)\} = \{r \cos \alpha, r \sin \alpha\}$ is therefore

$$z(t) = \begin{cases} \frac{r}{s} e^{i\alpha} \left( s \cosh\left(\frac{st}{2}\right) + \left( \sigma e^{i2(\theta - \alpha)} + i\zeta \right) \sinh\left(\frac{st}{2}\right) \right) & \text{if } \sigma^2 > \zeta^2 \\ \frac{r}{\bar{s}} e^{i\alpha} \left( \bar{s} \cos\left(\frac{\bar{s}t}{2}\right) + \left( \sigma e^{i2(\theta - \alpha)} + i\zeta \right) \sin\left(\frac{\bar{s}t}{2}\right) \right) & \text{if } \sigma^2 < \zeta^2 \end{cases} \tag{12}$$

and the associated velocity $w(t) = u(t) + iv(t)$ is given by

$$w(t) = \begin{cases} \frac{r}{2} e^{i\alpha} \left( s \sinh\left(\frac{st}{2}\right) + \left( \sigma e^{i2(\theta - \alpha)} + i\zeta \right) \cosh\left(\frac{st}{2}\right) \right) & \text{if } \sigma^2 > \zeta^2 \\ \frac{r}{2} e^{i\alpha} \left( -\bar{s} \sin\left(\frac{\bar{s}t}{2}\right) + \left( \sigma e^{i2(\theta - \alpha)} + i\zeta \right) \cos\left(\frac{\bar{s}t}{2}\right) \right) & \text{if } \sigma^2 < \zeta^2 \end{cases} \tag{13}$$

where we have defined the complementary Okubo–Weiss parameter by $\bar{s}^2 \equiv \zeta^2 - \sigma^2$. The mean-square distance of a particle from the origin is given by

$$\frac{1}{T} \int_0^T z(t)^2 \, dt = \begin{cases} \frac{2r^2}{Ts^3} \sinh\left(\frac{sT}{2}\right) \left[ \sigma A \cosh\left(\frac{sT}{2}\right) + sB \sinh\left(\frac{sT}{2}\right) \right] - \frac{r^2}{s^2} \zeta C & \text{if } \sigma^2 > \zeta^2 \\ \frac{2r^2}{T\bar{s}^3} \sin\left(\frac{\bar{s}T}{2}\right) \left[ -\sigma A \cos\left(\frac{\bar{s}T}{2}\right) + \bar{s}B \sin\left(\frac{\bar{s}T}{2}\right) \right] + \frac{r^2}{T\bar{s}^2} \zeta C & \text{if } \sigma^2 < \zeta^2 \end{cases} \tag{14}$$

and total velocity variance,

$$\frac{1}{T} \int_0^T w(t)^2 \, dt = \begin{cases} \frac{r^2}{2sT} \sinh\left(\frac{s}{2}T\right) \left[ \sigma A \cosh\left(\frac{s}{2}T\right) + sB \sinh\left(\frac{s}{2}T\right) \right] + \frac{r^2 \zeta C}{4} & \text{if } \sigma^2 > \zeta^2 \\ \frac{r^2}{2\bar{s}T} \sin\left(\frac{\bar{s}}{2}T\right) \left[ \sigma A \cos\left(\frac{\bar{s}}{2}T\right) - \bar{s}B \sin\left(\frac{\bar{s}}{2}T\right) \right] + \frac{r^2 \zeta C}{4} & \text{if } \sigma^2 < \zeta^2 \end{cases} \tag{15}$$

where

$$A = \sigma + \zeta \sin 2(\theta - \alpha), \quad B = \sigma \cos 2(\theta - \alpha), \quad C = \zeta + \sigma \sin 2(\theta - \alpha), \tag{16}$$

and $T$ is the length of time that has passed since the particle has moved from its initial position.

The Lagrangian frequency spectrum of a particle in a linear velocity field can now be computed using Equations (13) and (7) which yields

$$S(\omega) = \begin{cases} \frac{r^2}{T} \sinh^2\left(\frac{sT}{4}\right) \left[ \frac{\sigma A \cosh\left(\frac{s}{2}T\right) + sB \sinh\left(\frac{s}{2}T\right) - \zeta C}{\omega^2 + \frac{s^2}{4}} + \frac{s^2 C(\omega + \zeta/2)}{\left(\omega^2 + \frac{s^2}{4}\right)^2} \right] & \text{if } \sigma^2 > \zeta^2 \\ \frac{r^2}{T} \sin^2\left(\frac{\bar{s}T}{4}\right) \left[ \frac{-\sigma A \cos\frac{\bar{s}T}{2} + \bar{s}B \sin\frac{\bar{s}T}{2} + \zeta C}{\omega^2 - \frac{\bar{s}^2}{4}} + \frac{\bar{s}^2 C(\omega + \zeta/2)}{\left(\omega^2 - \frac{\bar{s}^2}{4}\right)^2} \right] & \text{if } \sigma^2 < \zeta^2 \end{cases} \tag{17}$$

where the Lagrangian frequency spectra of complex-valued velocities are permitted to be asymmetric in $\omega$ (see [18]), which will occur in Equation (17) when $\zeta \neq 0$. Asymmetric spectra arise when the rotary spectra are unequal and there is a preferred direction of spin [19]. With no strain and after sufficiently long observation time ($T \gg 1/\zeta$), Equation (17) becomes a single frequency delta function, reflecting the rotation of a particle from the vorticity. However, for the cases considered here, observation times are at most $O(1/s, 1/\bar{s})$, and often much less. The result is a spectrum that is generally very red ($S(\omega) \sim \omega^{-2}$), with total power increasing in observation time $T$.

The Lagrangian frequency spectrum in Equation (17) would appear to indicate that particles advected by a linear velocity field have a non-zero diffusivity, following the definition of Equation (8). However, while it is true that the linear velocity field causes particles to disperse, increasing their second moment with $T$, this spreading is entirely deterministic with correlations between particles spatially and across time, and thus does not formally meet the requirement that diffusivity results from a stationary random velocity process. From the perspective of trying to isolate and estimate the diffusivity of submesoscale processes (which may be stationary at these scales), the linear velocity field may be viewed as contaminating the lowest frequencies in the spectrum, providing erroneously high values of diffusivity if not removed correctly.

Figure 2 shows the one-sided Lagrangian frequency spectrum of a single particle simulated using Equation (9). The Lagrangian frequency spectrum thus has two distinguishing parts: the white noise submesoscale process given by Equation (10) and the deterministic red process given by Equation (17). In Figure 2 the observed particle spectrum is very nearly the linear addition of the theoretical Lagrangian frequency spectra of the mesoscale and submesoscale models of Equations (10) and (17) respectively. In terms of Figure 2, the objective of the methodology is to remove the deterministic contribution of the mesoscale flow (in blue), in order to study the submesoscale process that remains.

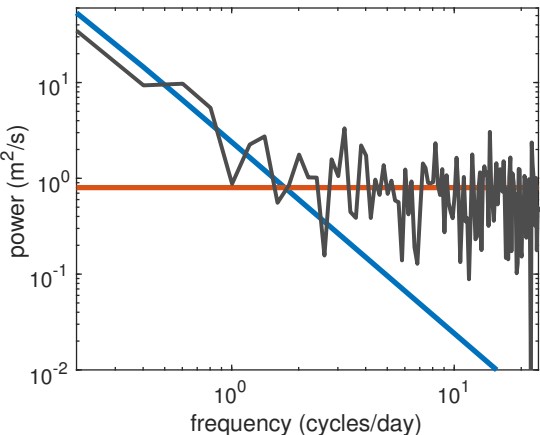

**Figure 2.** The one-sided frequency spectrum for a particle integrated with Equation (9) is shown in black. The particle is initially placed at $\{x(0), y(0)\} = \{1\,\text{km}, 1\,\text{km}\}$ and integrated for 5 days in a strain-only model with simulation parameters set to $\kappa = 0.1\,\text{m}^2/\text{s}$ and $\sigma = 1 \times 10^{-5}/\text{s}$. The theoretical spectrum of the mesoscale process, Equation (17), is shown in blue, and the theoretical spectrum of the white noise process, Equation (10), is shown in red.

## 3. Estimation and Hierarchical Modelling

The spreading of particles in the ocean can be categorised into three distinct stages of diffusivity according to the size of the drifter separation (or the tracer patch) relative to the size of mesoscale features [7]. At the smallest spatial scales, the mesoscale features may be so weak that the submesoscale processes dominate across all resolved scales and therefore completely control the spreading (e.g., when the mesoscale spectrum in Figure 2 is below the submesoscale spectrum). At the other extreme, where drifters are separated by distances that exceed the size of mesoscale features such as with the Global Drifter Program, the motions between any two drifters are uncorrelated and there are no common features to parameterise. We are interested in the middle stage, where the spread of the drifters is within the size of the mesoscale features. The upper bound of separation is dictated by the requirement that the spatial gradients in Equation (2) must be similar between drifters, while the lower bound is simply determined by lack of statistical significance of the mesoscale parameters. We place no upper bound on the number of drifters required, however there should be at least two drifters to remove the background part of the flow. The drifters should be sampled frequently enough that there is enough data to obtain estimates which are statistically significant whilst keeping the spread of the drifters within

the mesoscale. Further discussion of how to ensure significance of results will be given in Section 4 and Appendix A.

### 3.1. Parameter Estimation

Estimates for the mesoscale parameters in Equation (2) from observations will be obtained using least squares regression, by minimising the sum of the squared residuals representing the non-mesoscale flow, as we shall now show. This approach therefore fits as much of the data to the mesoscale part of the model as possible. To perform the fits, we make the important step of decomposing the $K$ drifter velocities into $K$ drifter velocities relative to the centre-of-mass, plus a centre-of-mass velocity, as represented in Equations (3) and (4) respectively. In other words the summation of Equations (3) and (4) recovers Equation (2). When put into matrix-vector notation for observations these models can be jointly written as

$$U = XA + \epsilon, \tag{18}$$

where we have defined

$$U = \frac{d}{dt} \underbrace{\begin{bmatrix} \bar{x}_k(t_n) \\ \bar{y}_k(t_n) \\ m_x(t_n) \\ m_y(t_n) \end{bmatrix}}_{2(K+1)N \times 1}, \quad \epsilon = \underbrace{\begin{bmatrix} u_k^{\text{sm}}(t_n) \\ v_k^{\text{sm}}(t_n) \\ u^{\text{bg}}(t_n) \\ v^{\text{bg}}(t_n) \end{bmatrix}}_{2(K+1)N \times 1}, \tag{19}$$

and

$$X = \frac{1}{2} \underbrace{\begin{bmatrix} \mathbf{0}_{KN} & \mathbf{0}_{KN} & \mathbf{0}_{KN} & \mathbf{0}_{KN} & \bar{x}_k(t_n) & \bar{y}_k(t_n) & -\bar{y}_k(t_n) & \bar{x}_k(t_n) \\ \mathbf{0}_{KN} & \mathbf{0}_{KN} & \mathbf{0}_{KN} & \mathbf{0}_{KN} & -\bar{y}_k(t_n) & \bar{x}_k(t_n) & \bar{x}_k(t_n) & \bar{y}_k(t_n) \\ 2 \cdot \mathbf{1}_N & \mathbf{0}_N & 2t_n & \mathbf{0}_N & \bar{m}_x(t_n) & \bar{m}_y(t_n) & -\bar{m}_y(t_n) & \bar{m}_x(t_n) \\ \mathbf{0}_N & 2 \cdot \mathbf{1}_N & \mathbf{0}_N & 2t_n & -\bar{m}_y(t_n) & \bar{m}_x(t_n) & \bar{m}_x(t_n) & \bar{m}_y(t_n) \end{bmatrix}}_{2(K+1)N \times p}, \quad A = \underbrace{\begin{bmatrix} u_0 \\ v_0 \\ u_1 \\ v_1 \\ \sigma_n \\ \sigma_s \\ \zeta \\ \delta \end{bmatrix}}_{p \times 1}. \tag{20}$$

In this notation, $\bar{x}_k(t_n) \equiv x_k(t_n) - m_x(t_n)$, $\bar{y}_k(t_n) \equiv y_k(t_n) - m_y(t_n)$ are length $KN$ column vectors of the $N$ observations at times $t_1 \leq t_n \leq t_N$ from each of the $K$ drifters in a chosen time window of width $W = t_N - t_1$. Similarly $\bar{m}_x(t_n) \equiv m_x(t_n) - x_0$, $\bar{m}_y(t_n) \equiv m_y(t_n) - y_0$ are length $N$ column vectors of the moving centre-of-mass at times $t_1 \leq t_n \leq t_N$. The particular ordering of the observations within each vector in Equations (19) and (20) does not matter, so long as it is consistent, and in fact, there is no restriction that the drifter observations occur at the same time, despite our choice of notation. We have defined $\mathbf{0}_{KN}$ and $\mathbf{1}_{KN}$ to be $KN \times 1$ column vectors of zeros and ones, respectively. Under each matrix we have given its size, where $p$ is the number of parameters, and in this case $p = 8$. The vector $A$ contains model parameters which are estimated using the least squares solution

$$A = (X'X)^{-1}X'U. \tag{21}$$

By combining Equations (18) and (21) the residual submesoscale and background velocities can be estimated by taking

$$\epsilon = [1 - X(X'X)^{-1}X']U. \tag{22}$$

The least-squares solution is equivalent to the optimal maximum likelihood solution when the residuals are Gaussian and independent and identically distributed. In general,

weighted least squares solutions should be used if residuals are correlated or have unequal variance, and although this will likely be the case here, weighted-least squares requires prior knowledge of the distributional structure of the residuals which we do not wish to assume is known. Overall, we found the (non-weighted) least squares solution of Equations (21)–(22) to be robust in simulation experiments and real data analysis, and to perform better than performing least squares directly on the representation of Equation (2) on raw velocities for each drifter without removing centre-of-mass. This is due to the fact that the $K$ drifter velocities in centre-of-mass coordinates, with the addition of the centre-of-mass velocity, can be thought of as a collection of $K + 1$ drifters that are more independent of each other than the $K$ drifters in fixed-reference frame coordinates. This leads to errors that are more uncorrelated over drifters yielding better least squares parameter fits.

### 3.2. Flow Decomposition

Once the parameters have been estimated using Equation (21), the constituent parts of the conceptual model of Equation (1) can be computed. The mesoscale contribution to each drifter is computed using

$$
\begin{bmatrix} u_k^{\mathrm{meso}}(t_n) \\ v_k^{\mathrm{meso}}(t_n) \end{bmatrix} \equiv \begin{bmatrix} u_0 + u_1 t \\ v_0 + v_1 t \end{bmatrix} + \frac{1}{2} \begin{bmatrix} \sigma_n + \delta & \sigma_s - \zeta \\ \sigma_s + \zeta & \delta - \sigma_n \end{bmatrix} \begin{bmatrix} x_k(t_n) - x_0 \\ y_k(t_n) - y_0 \end{bmatrix}. \tag{23}
$$

The background is assumed to be spatially homogeneous, and thus can be recovered from the residuals by averaging across drifters at each time,

$$
\begin{bmatrix} u^{\mathrm{bg}}(t_n) \\ v^{\mathrm{bg}}(t_n) \end{bmatrix} \equiv \frac{1}{K} \sum_{k=1}^{K} \left( \frac{d}{dt} \begin{bmatrix} x_k(t_n) \\ y_k(t_n) \end{bmatrix} - \begin{bmatrix} u_k^{\mathrm{meso}}(t_n) \\ v_k^{\mathrm{meso}}(t_n) \end{bmatrix} \right). \tag{24}
$$

Finally, the submesoscale contribution to each drifter is all that remains,

$$
\begin{bmatrix} u_k^{\mathrm{sm}}(t_n) \\ v_k^{\mathrm{sm}}(t_n) \end{bmatrix} \equiv \frac{d}{dt} \begin{bmatrix} x_k(t_n) \\ y_k(t_n) \end{bmatrix} - \begin{bmatrix} u_k^{\mathrm{meso}}(t_n) \\ v_k^{\mathrm{meso}}(t_n) \end{bmatrix} - \begin{bmatrix} u^{\mathrm{bg}}(t_n) \\ v^{\mathrm{bg}}(t_n) \end{bmatrix}. \tag{25}
$$

This accomplishes the conceptual decomposition of velocities proposed in Equation (1). We emphasise that the fits of Equations (18)–(22) could be performed without the centre-of-mass velocity by removing the bottom two rows of $U$, $\epsilon$ and $X$ in Equations (19) and (20). This is in effect only fitting observations to the second-moment model of Equation (4), as also proposed in [12]. While this fit still obtains estimates of mesoscale quantities $\{\sigma, \theta, \zeta, \delta\}$, and disentangles the submesoscale $\{u^{\mathrm{sm}}(t), v^{\mathrm{sm}}(t)\}$, the first-moment mesoscale parameters $\{u_0, u_1, v_0, v_1\}$ and the background $\{u^{\mathrm{bg}}, v^{\mathrm{bg}}\}$ can no longer be estimated directly (unless fitted a posteriori). This means a full decomposition of the flow as performed in Equations (23)–(25) is not directly accomplished using the $K$ drifters in centre-of-mass frame only. We shall refer to this reduced technique as the second-moment fitting method. In contrast, we refer to the full estimation technique from Equations (18)–(25) as the first and second-moment fitting method.

Regardless of the fitting method, we estimate the isotropic submesoscale diffusivity $\kappa_{k,z}^{\mathrm{sm}}(t)$, defined in Equation (6), by measuring the implied square displacement of the submesoscale velocities within the window. This yields

$$
\widehat{\kappa}_{k,z}^{\mathrm{sm}}(t_n) = \frac{\Delta}{4N} \left| \sum_{t=t_1}^{t_N} u_k^{\mathrm{sm}}(t) + i v_k^{\mathrm{sm}}(t) \right|^2, \tag{26}
$$

where $\Delta$ is the sampling interval of drifter observations measured in seconds. Equation (26) is equivalent to taking $1/4$ of the periodogram of the velocities—or the absolute square of the Fourier Transform—at frequency zero. This is consistent with the fact that the theoretical diffusivity of a stationary complex-valued process is determined by $1/4$ of the zero-frequency of the Lagrangian frequency spectrum as per Equation (8).

The above equations produce estimates of the background, mesoscale and submesoscale parts of the flow over some choice of temporal window length $W = t_N - t_1$. A small value of $W$ results in a reduced number of data points in the regression causing potentially noisy parameter estimates. Conversely, a large value of $W$ incorporates more distant observations in time and smooths over this noise, but may lead to poor estimates if the underlying mesoscale parameters are evolving over time. This is the classic bias-variance trade-off in statistical estimation. In Section 4, we address the issue of choosing an appropriate window length, and we introduce a principled estimation method using splines that allow parameters to evolve slowly over time, resulting in smoother less-variable estimates.

### 3.3. Hierarchical Modelling

The Taylor series model of Equation (2) specifies eight mesoscale parameters, specified by $\{u_0, v_0, u_1, v_1, \sigma, \theta, \zeta, \delta\}$, and these can be estimated from clustered drifter data using the machinery of Section 3.1. However, not every clustered set of drifters will necessarily experience all of these effects (as we illustrated in Figure 1), or the data might not give statistically significant estimates of some of the parameters even if they are truly present. Alternatively, we might already know the true values of some of the parameters and so we do not wish to estimate these. Motivated by this, we now introduce a simple method of removing certain parameters from the model, by either setting them to be zero or a pre-specified fixed value, and then estimating only the remaining unspecified parameters. If we were to instead set parameters to zero (or fixed values) after estimation, we would sub-optimally lose part of the data contained in the removed estimate.

To remove a parameter from the model, one simply removes the parameter from the vector $A$ in Equation (20) and the corresponding column from the matrix $X$. In a similar vein, multiple parameters can be removed by repeating this procedure. Ultimately, depending on the number of parameters removed, the matrix $X$ will be sized $2(K + 1)N \times p$, and the column vector $A$, will be sized $p \times 1$, where $p$ is the number of free parameters that remain in the model. If $p = 8$, as presented in Equation (19), then this represents the full mesoscale solution. If any parameter values are known a priori then they should be inserted as fixed values into $A$ and then multiplied by the corresponding respective columns from $X$ and then subtracted from the vector $U$, before proceeding with the least squares minimisation of Equation (21) to estimate remaining parameters.

We now consider the special case of only estimating the mesoscale quantities $\{\sigma, \theta, \zeta, \delta\}$ using the second-moment fitting method discussed in Section 3.2. If we estimate all quantities in $\{\sigma, \theta, \zeta, \delta\}$ then $p = 4$. In contrast, if we remove all mesoscale parameters such that $\{\sigma, \theta, \zeta, \delta\} = \{0, 0, 0, 0\}$, then $p = 0$, and only submesoscale velocities remain in the centre-of-mass frame of Equation (4). If $0 < p < 4$, this represents scenarios where some mesoscale components from $\{\sigma, \theta, \zeta, \delta\}$ are present, and some are not, and we display this schematically in Figure 3. We consider strain rate and strain angle (or equivalently shear and normal strain rates) to be either jointly present or both missing. Overall, there are therefore eight possible models we might consider, shown explicitly in Figure 3. Regardless of the choice of model, the remaining non-zero parameters are estimated using Equation (21) as before.

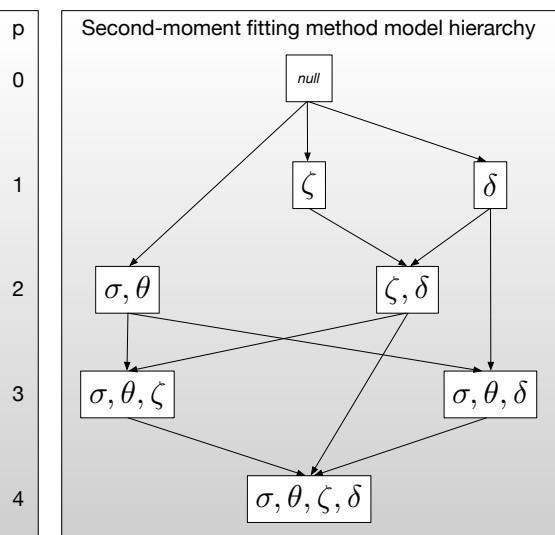

**Figure 3.** Hierarchy of mesoscale models using the second-moment fitting method where *p* indicates the number of parameters. A model with increased complexity is used only if it explains significantly more variance than the lower complexity model. Models with fewer parameters are favoured when a choice must be made.

Figure 3 also shows that the eight models exist in a hierarchy. The simplest model, the null hypothesis shown at the top of Figure 3, corresponds to velocities in a centre-of-mass frame that are submesoscale only. There are three direct descendants of this model in the hierarchy, the addition of vorticity or divergence, each of which requires one more parameter, or strain, which requires two additional parameters. The central philosophy is that a descendent in the hierarchy should only be used if it shows meaningful improvement in some relevant error metric, essentially disproving the null hypothesis. Because adding parameters will always produce at most the same residual (which may itself be the error metric), this approach avoids using too many degrees-of-freedom and producing meaningless or noisy parameter estimates.

It is worth noting that estimating all four mesoscale parameters $\{\sigma, \theta, \zeta, \delta\}$ at each time point (as is often done in the literature) would benefit from this conceptual approach. With $K$ drifters there are $2K$ position observations at a given time point, from which four parameters must be estimated at each time point. For modestly sized drifter deployments, this computation runs the risk of producing estimates with no statistical significance.

In general, when selecting between the model hierarchies for all eight mesoscale parameters $\{u_0, v_0, u_1, v_1, \sigma, \theta, \zeta, \delta\}$ then we are faced with an increased complexity of selecting between reduced permutations of the full specification. Motivated by this, in Section 4.3 we will introduce methodology for estimating time-varying parameters using splines, which allows for a natural mechanism from which to build a full hierarchy of first and second-moment candidate models, as we shall show.

### 3.4. Selecting between Hierarchies

We have provided a mixed background-mesoscale-submesoscale modelling framework in Equation (2) and a corresponding estimation framework in Section 3.1. Then in Section 3.3 we discussed how to estimate parameters using different hierarchies of mesoscale components in the overall model. The appropriateness of a chosen model in the hierarchy, for a given set of observational drifter data, can be evaluated by estimating the error resulting from the fitted model at a given point in time. We argue there is more than one meaningful way in which error can be computed—and in this section we shall define two such ways that prove to be very useful in terms of model evaluation.

3.4.1. Fraction of Variance Unexplained (FVU)

The first method is perhaps the most intuitive. Here we calculate how much variance remains in the 'unexplained' residual submesoscale velocities found in Equation (25). This value in itself, however, is not a meaningful quantity unless it is presented in reference to some other quantity. Therefore, to provide a normalised and meaningful metric we introduce the notion of the Fraction of Variance Unexplained (FVU), which is defined as

$$
\text{FVU} = \frac{\sum_{t_n=t_1}^{t_N} \sum_{k=1}^{K} \left\{ u_k^{\text{sm}}(t_n)^2 + v_k^{\text{sm}}(t_n)^2 \right\}}{\sum_{t_n=t_1}^{t_N} \sum_{k=1}^{K} \left\{ \left[ \frac{d}{dt} \left( x_k(t_n) - m_x(t_n) \right) \right]^2 + \left[ \frac{d}{dt} \left( y_k(t_n) - m_y(t_n) \right) \right]^2 \right\}},
\tag{27}
$$

and hence quantifies the proportion of the variability remaining in the submesoscale model, as compared to velocities that have only had the centre-of-mass removed (and will hence still contain second-moment mesoscale effects present in Equation (4)). The FVU will therefore in general be some value between zero and one. An FVU value close to one occurs when there is little to no mesoscale component estimated from the data. In contrast, an FVU value equal to zero means the mesoscale model successfully explains all variability in the data after the background is removed, and there is no residual submesoscale process left behind. For mixed mesoscale and submesoscale flow the FVU will be somewhere between zero and one, and this will vary dependent on the magnitude and number of mesoscale components present in the model fit.

In Figure 4, in the left column we display FVU values obtained from our simulation setup shown in Figure 1. Specifically, we generate 100 replicated simulations of each of the four model scenarios shown in Figure 1—diffusivity only, strain+diffusivity, strain+vorticity+diffusivity (strain dominated), strain+vorticity+diffusivity (vorticity dominated) —where the stochasticity between replicates occurs from simulating submesoscale velocities from a Gaussian white noise process as in Equation (9). Again, as in LatMix Site 1, we simulate nine drifters within each simulation with matching initial positions, but this time we just simulate half-hourly records for one day. We use the procedures described in Section 3.3 to fit four hierarchies of models to each simulation within each scenario. Note that we perform a global fit by setting the window length *W* to be the full length of the observations (one day). The FVU values are calculated from Equation (27) and the resulting spread of values across simulations are shown by box and whisker plots in Figure 4. We also provide the spread of observed FVU values in an oracle case where the true mesoscale parameters are known.

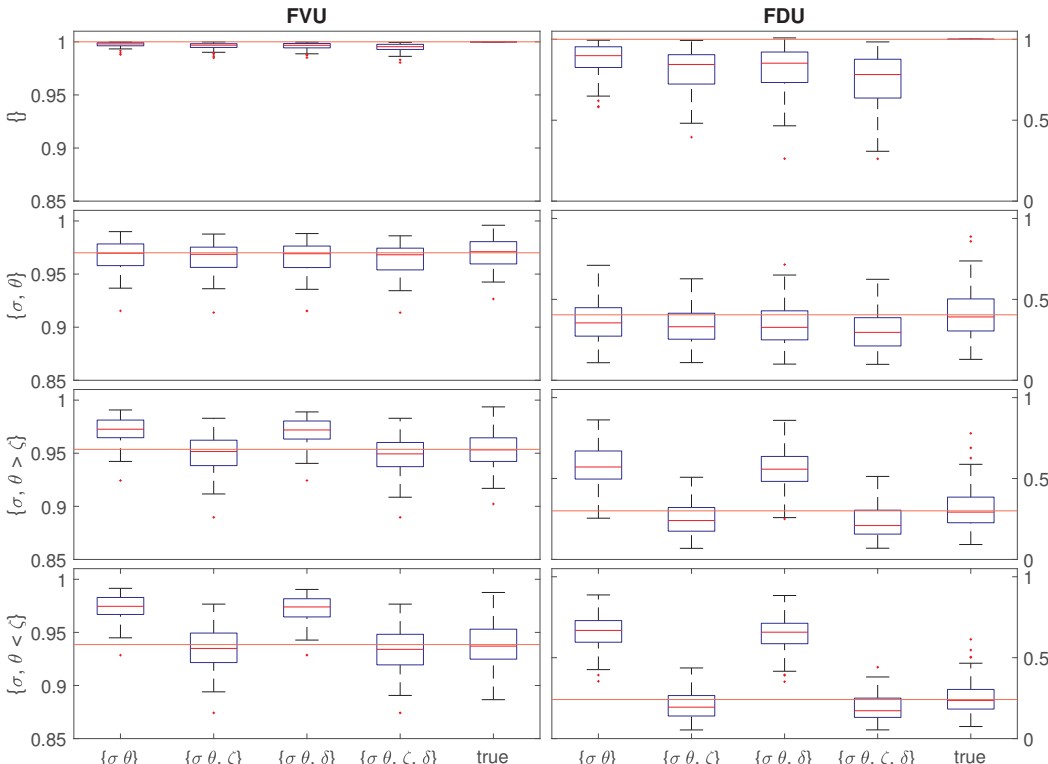

**Figure 4.** FVU (left column) and FDU (right column) for candidate models fitted to trajectories generated from the four model scenarios from Figure 1. Each subplot here is for a different true model scenario (the *y*-axis), and each box and whisker within a subplot provides the spread of FVU/FDU values from a fitted candidate model (the *x*-axis). The final box and whisker in each subplot is using the true mesoscale parameter values. The spread of results is over 100 repeated simulations using nine drifters sampled every 30 min for one day. The estimated theoretical FVU, obtained from Equation (28), and the estimated theoretical FDU, obtained from Equation (30), are overlaid by a red horizontal line in each subplot. Parameters are estimated using the second-moment fitting method, where results using the first and second-moment fitting method yield near identical results as $u_0 = v_0 = u_1 = v_1 = u^{\mathrm{bg}} = v^{\mathrm{bg}} = 0$ in these simulations.

In the figure we have also indicated the estimated theoretical FVU value obtained by combining the mesoscale variance obtained from Equation (15) for each drifter $k$ (let us denote this $\sigma^2_{w^{\mathrm{meso}}}(k)$) with the submesoscale variance of a white noise process given from the spectral form of Equation (10) yielding $\sigma^2_{w^{\mathrm{sm}}} = 4\kappa(1 - 1/K)$, which is the same for each drifter, where the $(1 - 1/K)$ rescaling is required to account for moving to a centre-of-mass reference frame. We can then obtain an estimated theoretical FVU value, which we denote $\widetilde{\mathrm{FVU}}$, by taking

$$\widetilde{\mathrm{FVU}} = \frac{\sigma^2_{w^{\mathrm{sm}}}}{\left\{ \frac{1}{K} \sum_{k=1}^{K} \sigma^2_{w^{\mathrm{meso}}}(k) \right\} + \sigma^2_{w^{\mathrm{sm}}}}. \tag{28}$$

This an estimated theoretical FVU, rather than an exact solution, because we have ignored the co-dependence between the mesoscale and submesoscale processes and assumed these variances aggregate separately. The results however indicate remarkable agreement between theoretical and observed quantities for FVU over all scenarios (except when insufficient mesoscale parameters are proposed in the candidate model), suggesting Equation (28) is an accurate approximation for the spatial and temporal scale of the simulation performed.

Overall, the key finding of Figure 4 (left column) is that the FVU helps identify the correct model in all true model scenarios considered, and correctly estimates how much of the variance is explained by the mesoscale and submesoscale components in

agreement with the theory. The addition of a mesoscale parameter which is truly present significantly reduces the FVU, but adding further unnecessary mesoscale parameters (such as the divergence which is not present in any of the scenarios) does not significantly reduce FVU. This diagnostic tool therefore shows utility as a method for detecting the presence of mesoscale effects on drifter velocities, and for selecting between mesoscale model hierarchies. We shall scrutinise this further when we apply our procedures to LatMix data in Section 5.

3.4.2. Fraction of Diffusivity Unexplained (FDU)

The FVU is a measure of how much of the variability of the data remains in the submesoscale residuals. However, we argue this is not the only metric with which to ultimately select from a model hierarchy. First of all, as the residual velocities are being directly minimised (along with the background) in the least squares fits of Equations (18)–(22), the more complex models will generally have a lower FVU than nested simpler models with fewer or no mesoscale components (as seen in Figure 4). This may lead to over-fitting models unless parameter penalisation methods are introduced. Secondly, mesoscale processes are primarily low frequency processes with decaying Lagrangian velocity frequency spectra, as we showed in Figure 2. Submesoscale processes, on the other hand, will likely have Lagrangian velocity frequency spectra that are spread across frequencies and concentrated away from frequency zero. For example, white noise submesoscale residuals will have a flat spectrum, and an internal wave process, represented by the Garrett–Munk spectrum for instance, will have significant energy at the inertial frequency $f_0$, but very small energy at frequency zero.

For these reasons, we now motivate a second metric with which to evaluate different model hierarchies. Specifically, we measure the diffusivity of the residual process for each drifter, and compare this with the implied total diffusivity of each drifter when no mesoscale is removed. In other words, we compare the variability of the aggregated and submesoscale-only components in terms of their respective diffusivities, with a view that submesoscale diffusivity should be much lower than total diffusivity when even a mild mesoscale component is present (as mesoscale energy is dominant at low frequencies in the velocity spectra). To quantify this effect we introduce the notion of the Fraction of Diffusivity Unexplained (FDU), which we define by

$$\text{FDU} = \frac{\sum_{t_n=t_1}^{t_N} \sum_{k=1}^{K} \hat{\kappa}_{k,z}^{\text{sm}}(t_n)}{\sum_{t_n=t_1}^{t_N} \sum_{k=1}^{K} \hat{\kappa}_{k,z}^{\text{c.o.m.}}(t_n)}, \tag{29}$$

where $\hat{\kappa}_{k,z}^{\text{sm}}(t_n)$ has already been defined in Equation (26). $\hat{\kappa}_{k,z}^{\text{c.o.m.}}(t_n)$ is the diffusivity for drifter $k$ with only centre-of-mass removed, which is defined by replacing $u_k^{\text{sm}}(t)$ with $\frac{d}{dt}(x_k(t_n) - m_x(t_n))$ and $v_k^{\text{sm}}(t)$ with $\frac{d}{dt}(y_k(t_n) - m_y(t_n))$ in Equation (26). The FDU measures how much diffusivity is present in the submesoscale residual after removing the mesoscale, as compared to the diffusivity that is observed relative to the centre-of-mass when no mesoscale has been explicitly removed. An FDU value of zero means that the submesoscale process has no observed diffusivity, and an FDU of one will occur when either no mesoscale is present, or the mesoscale does not create any diffusive-type behaviour on the particles.

We display observed FDU values across our simulations in the right column of Figure 4, mirroring the simulation setup used for FVU described in Section 3.4.1. The estimated theoretical FDU values are overlaid by a red horizontal value from computing

$$\widetilde{\text{FDU}} = \frac{\kappa_z^{\text{sm}}}{\left\{\frac{1}{K}\sum_{k=1}^{K} \kappa_{k,z}^{\text{meso}}\right\} + \kappa_z^{\text{sm}}}, \tag{30}$$

where the expected submesoscale diffusivity for all drifters is $\kappa_z^{\text{sm}} = \kappa(1-1/K)$ where again the $(1-1/K)$ rescaling is required to account for moving to a centre-of-mass reference

frame. We obtain $\kappa_{k,z}^{\mathrm{meso}}$ by taking 1/4 of the zero-frequency value from Equation (17) (as per the definition of Equation (8)). Similarly to Equation (28), Equation (30) is an estimated theoretical FDU because we are assuming independent dispersion caused by the mesoscale and submesoscale. Nevertheless, Figure 4 indicates consistent agreement between observed and theoretical FDU values (when the correct model is fitted), highlighting the accuracy of this approximation.

The main finding of the FDU analysis in Figure 4 is that the mesoscale explains significantly more of the total diffusivity than the total variance. This is as expected because of the low-frequency nature of mesoscale processes (see Figure 2) and highlights the usefulness of computing FDU values to test for mesoscale presence. In all cases we can see that FDU analysis reveals the correct generating mesoscale model even better than FVU does. We shall further use this diagnostic method of assessing model fits with LatMix data in Section 5.

## 4. Uncertainty Quantification and Capturing Temporal Evolution

### 4.1. Uncertainty Quantification

We now provide a method for estimating the uncertainty of parameter estimates when applied to observational datasets. In a simulation setting, uncertainty estimates can be obtained by repeating experiments several times stochastically or with different initial conditions, but this cannot be done in the real world where clustered drifter deployments are scarcely repeated in the same region of the ocean, and will likely be measuring different mesoscale and submesoscale features each time.

Instead, we resort to the bootstrap, which resamples the observed data in such a way as to provide a population of different datasets with which to measure uncertainty. Specifically, the bootstrap is implemented by taking a random sample of $K$ trajectories from the $K$ drifters with replacement, such that the same trajectories may be selected multiple times as if they were different drifters. Then the mesoscale parameters are estimated for this random sample of trajectories. Let us denote any one of these parameter estimates as $\hat{p}_b$. The process is then repeated $B$ times, every time randomly resampling a set of $K$ trajectories with replacement, such that we obtain $B$ parameter estimates $\{\hat{p}_1, \ldots, \hat{p}_B\}$. These replicated bootstraps can be used to form quantiles which then provide confidence intervals for the parameter of interest, often set to values such as 90% or 95%. Alternatively, we can also estimate the standard error of $\hat{p}$, the parameter estimate for $p$, by measuring the sample standard deviation of $\hat{p}_b$ given by

$$\mathrm{SE}_B(\hat{p}) = \left[ \frac{1}{B-1} \sum_{i=1}^{B} \left\{ \hat{p}(i) - \hat{p}_{(\cdot)} \right\}^2 \right]^{1/2}, \tag{31}$$

where $\hat{p}_{(\cdot)} = \frac{1}{B} \sum_{i=1}^{B} \hat{p}(i)$.

In Figure 5 we show a histogram of bootstrap parameter estimates for $\{\sigma, \theta, \zeta\}$, with a red vertical line at the true value, and a blue vertical line showing the average bootstrap estimate. The purpose of this simulation is simply to show that bootstrap parameter estimates are centred at their true values and symmetrically distributed, despite the fact that drifter trajectories are sampled with replacement. We found this to be a consistent feature across different true parameter values and simulation settings.

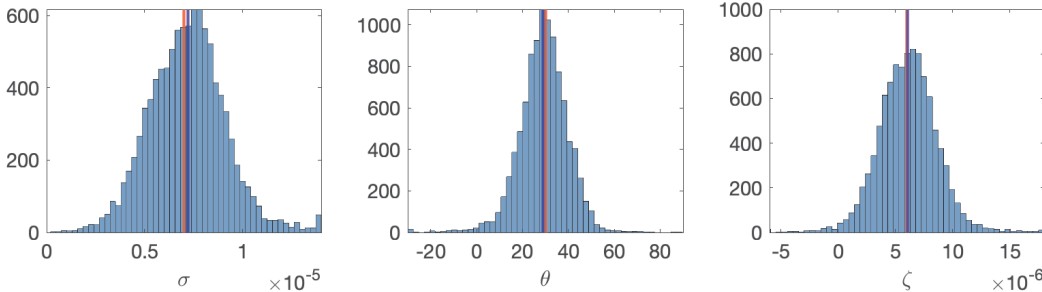

**Figure 5.** Histogram of bootstrap parameter estimates for strain rate, strain angle, and vorticity, over 100 repeated simulations where $B = 100$ for each simulation, thus obtaining 10,000 total bootstrapped parameter values. The trajectories are generated as in Figure 1 in the strain-dominated model for 1 day, and the parameters are estimated using the second-moment fitting method. Any bootstrap estimates outside the range of the *x*-axis are placed in the limiting visible bar in the histogram on each side. The red vertical line is the true parameter value, and the blue vertical line is the average bootstrap estimate.

Next we establish that the bootstrap estimate for the standard error of parameter estimates, given in Equation (31), agrees with standard errors of parameter estimates observed from repeated simulations. In Table 1 we compare simulated and bootstrap standard errors for two experiments: the strain-only and the strain-dominated simulations of Figure 1. The standard errors from simulations are across 100 repeated simulations, but the bootstrap standard error approximation is just from 1 simulation of drifters each time (as we would have with real data). Despite this, the average bootstrap standard error estimate is very close to the standard error from repeated simulations (with the standard deviation of the bootstrap standard error accounting for any difference). Notice also that the bootstrap standard error estimates are usually conservative, which is better than the converse, and correctly increase when more parameters need to be estimated. This demonstrates the accuracy of Equation (31) in estimating the standard error of parameter estimates obtained from Equation (21). We will make use of the bootstrap in the analysis of LatMix data in Section 5.

**Table 1.** Observed standard errors from simulation, and average bootstrap standard error estimates from Equation (31) (where $B = 100$), over 100 repeated simulations, for both the strain-only and strain-dominated simulations of Figure 1 over 1 day. We also provide the standard deviation of bootstrap standard error estimates across the 100 simulations, as indicated after the $\pm$ symbol.

| | $\sigma \ (s^{-1}) \times 10^6$ | $\theta \ (°)$ | $\zeta(s^{-1}) \times 10^6$ |
|---|---|---|---|
| Strain-only Simulation | | | |
| Simulated | 1.17 | 6.68 | N/A |
| Bootstrap | $1.32 \pm 0.365$ | $6.29 \pm 2.47$ | N/A |
| Strain-dominated Simulation | | | |
| Simulated | 1.22 | 6.78 | 1.61 |
| Bootstrap | $1.55 \pm 0.459$ | $8.08 \pm 3.43$ | $1.94 \pm 0.572$ |

*4.2. Time-Evolving Parameters Using Rolling Windows*

To estimate the temporal evolution of mesoscale features across a drifter deployment we allow the mesoscale parameters to evolve over time. In this section we first introduce a simple method for doing so where we use a rolling time window of width $W$ and estimate the parameters $\{u_0(t_n), v_0(t_n), u_1(t_n), v_1(t_n), \delta(t_n), \zeta(t_n), \sigma_n(t_n), \sigma_s(t_n)\}$ in Equation (20) over time using velocity observations contained in the interval $[\frac{d}{dt}x_k(t_n - \frac{W}{2}), \frac{d}{dt}x_k(t_n + \frac{W}{2})]$

and $[\frac{d}{dt}y_k(t_n - \frac{W}{2}), \frac{d}{dt}y_k(t_n + \frac{W}{2})]$ using the exact approach outlined in Section 3.1, repeated at every observation time-step $t_n$ in the experiment.

In general, the window width parameter $W$ should be chosen to be large enough to ensure we have reduced variance and statistically significant estimates of each mesoscale parameter, but not so large that resolution is lost from over-smoothing. To examine this effect we display simulated trajectories in Figure 6 which exactly follows the strain-only simulation from Figure 1, except that the strain rate parameter now decreases linearly by a factor of 10 across the length of the 6.25 day simulation, and we have increased $\kappa$ to $0.5\text{m}^2/\text{s}$. We then use the second-moment fitting method with the strain-only model over rolling windows with three choices of $W$ (6-hours, 1-day, or 3-days). In Figure 7 we display the time-varying strain rate estimate over time from the data in Figure 6, alongside the standard error of this estimate over time (obtained over 100 repeated simulations). With this increased diffusivity, the inherent trade-off with the rolling-window method becomes apparent. Long window lengths provide low uncertainty, but the parameter estimates are only provided in the temporal centre of the experiment (and would be biased if extended outwards). Short windows, on the other hand, provide variable estimates with large standard errors that exceed half the parameter value, as we see on the right panel—meaning such estimates cannot be statistically distinguished from zero in a "two sigma" sense. A daily window length is perhaps the most appropriate balance here.

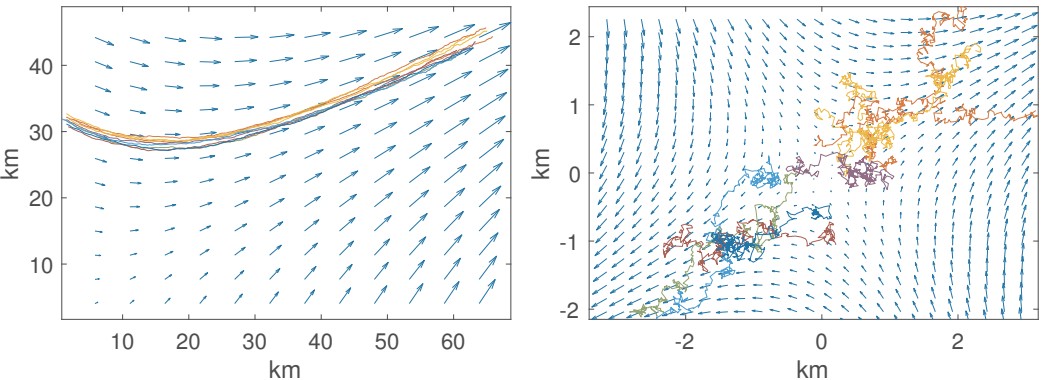

**Figure 6.** Simulation of nine drifters using the identical configuration of Figure 1 (strain only) except that the strain rate changes linearly across time from $\sigma = 1 \times 10^{-5}/\text{s}$ to $\sigma = 1 \times 10^{-6}/\text{s}$ and $\kappa = 0.5\text{m}^2/\text{s}$. The left panel displays drifter positions. The right panel displays drifter positions with respect to their centre-of-mass. The quiver arrows indicate the velocity field at the beginning of the simulation.

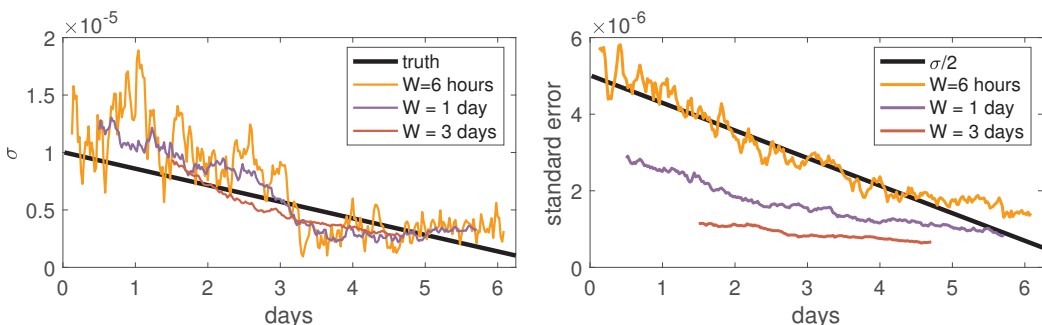

**Figure 7.** The left panel shows rolling-time window estimates of the varying strain rate from the data presented in Figure 6 over three choices of window lengths using the second-moment fitting method. The right panel shows the standard error of these time-varying estimates over 100 repeated simulations, plotted against the true value of $\sigma/2$.

Motivated by these challenges, we shall shortly provide a more principled approach to generating smoothly-evolving parameter estimates using splines in Section 4.3. Before doing so, we present results of a large simulation analysis which we will use to guide our

window length selection choices in the LatMix experiment. Specifically, in Figure 8 we plot a heatmap of standard errors in strain rate estimation, over a grid of values of true constant strain rate, $\sigma$, and estimation window length, $W$. We repeat the analysis for a low diffusivity $\kappa = 0.1$ m$^2$/s and high diffusivity setting $\kappa = 1$ m$^2$/s. Otherwise the settings are the LatMix-type settings used in Figure 1, using nine drifter trajectories with matching starting locations. The standard errors are in units of the true strain rate, and we have marked with a red line the point at which the standard error is approximately equal to half the true strain rate. The way in which this plot should be interpreted is that for a given strain rate (and diffusivity), the window length should be at least as long as the red line marking the point at which estimates become statistically significant. For example, higher diffusivities, or lower strain rates, will require longer windows with which to estimate the parameters significantly. We focus on strain in these simulations, as this was found to be the most pronounced mesoscale effect in the LatMix analysis that follows, but this analysis could be repeated with other mesoscale parameters to inform window length selection for other drifter deployments. In Appendix A we perform a brief sensitivity analysis of these results for varying numbers of drifters and initial deployment configurations, to help generalise our findings to wider settings.

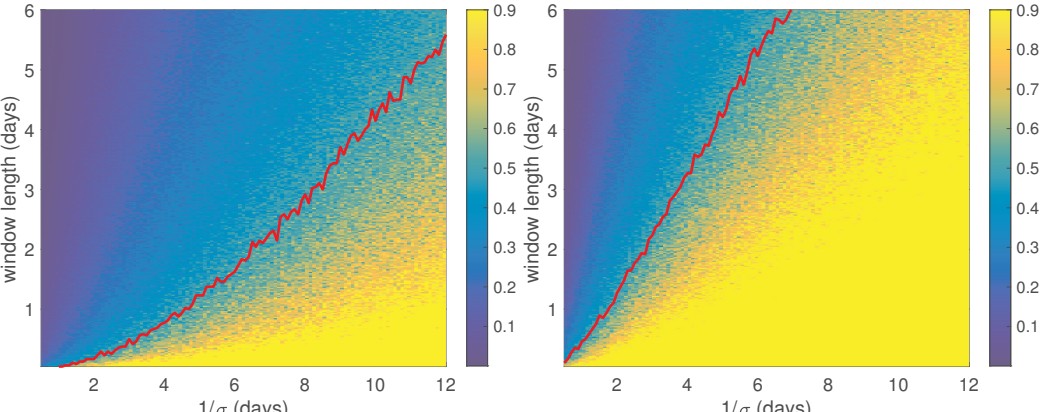

**Figure 8.** Estimated standard errors for the strain rate (in the units of the true strain rate) across a dense grid of fixed strain rate values $\sigma$ and window lengths $W$ in a strain-only simulation mirroring the setup in Figure 1. In the left panel we have set $\kappa = 0.1$ m$^2$/s and in the right $\kappa = 1$ m$^2$/s. The strain rate estimates are obtained using the second-moment fitting method of a strain-only model, and the standard errors are obtained over 100 repeated simulations. The standard errors in the heatmap are upper-bounded by 0.9 for representation purposes. We draw a red line where the standard error is approximated to be half the true parameter value for each value of the strain rate.

### 4.3. Slowly-Evolving Parameters Using Splines

To generalise the idea of time windowing to estimate the mesoscale parameters, we represent the parameters as coefficients as a finite sum of B-splines,

$$\sigma(t) = \sum_{m=1}^{M} \hat{\sigma}^m B^m(t), \tag{32}$$

where $M$ is the total number splines over the experiment window and $\hat{\sigma}^m$ are the $M$ coefficients. A B-spline (or basis spline) of degree $S$ is a local piecewise polynomial that maintains nonzero continuity across $S$ knot points placed at times $\tau_i$. These knot points define the extent of the B-splines, and therefore let us choose an effective window length for parameter fluctuations. The lowest degree ($S = 0$) splines are boxcar functions between the knot points, and are thus identical to non-overlapping windows in Section 4.2. At degree $S = 1$, B-splines are triangle functions that span two knot points, thus providing continuity in time as well as a piecewise first derivative. This generalises to higher degrees, where a B-spline of degree $S$ has $S$ non-zero derivatives, as reviewed in [11]. The key benefit to this

approach is that we can allow for time variation in the parameters while simultaneously choosing an effective window length—all while adding only a few coefficients to the model.

To extend the estimation method presented in Section 3.1, we now require $M$ coefficients for each of the $p$ parameters, resulting in $pM$ total coefficients to estimate. Rewriting vector $A$ from Equation (20) we have that

$$A = \underbrace{\begin{bmatrix} u_0^m \\ v_0^m \\ \sigma_n^m \\ \sigma_s^m \\ \zeta^m \\ \delta^m \end{bmatrix}}_{pM \times 1}, \tag{33}$$

where each coefficient, e.g., $u_0^m$, is a column vector of the $M$ B-spline coefficients (we will shortly discuss why $u_1$ and $v_1$ can be dropped here). The data matrix $X$ correspondingly expands from $p$ to $pM$ columns,

$$X = \frac{1}{2}\underbrace{\begin{bmatrix} \mathbf{0}_{KN} & \mathbf{0}_{KN} & \bar{x}_k(t_n)B^m(t_n) & \bar{y}_k(t_n)B^m(t_n) & -\bar{y}_k(t_n)B^m(t_n) & \bar{x}_k(t_n)B^m(t_n) \\ \mathbf{0}_{KN} & \mathbf{0}_{KN} & -\bar{y}_k(t_n)B^m(t_n) & \bar{x}_k(t_n)B^m(t_n) & \bar{x}_k(t_n)B^m(t_n) & \bar{y}_k(t_n)B^m(t_n) \\ 2B^m(t_n) & \mathbf{0}_N & \bar{m}_x(t_n)B^m(t_n) & \bar{m}_y(t_n)B^m(t_n) & -\bar{m}_y(t_n)B^m(t_n) & \bar{m}_x(t_n)B^m(t_n) \\ \mathbf{0}_N & 2B^m(t_n) & -\bar{m}_y(t_n)B^m(t_n) & \bar{m}_x(t_n)B^m(t_n) & \bar{m}_x(t_n)B^m(t_n) & \bar{m}_y(t_n)B^m(t_n) \end{bmatrix}}_{2(K+1)N \times pM}, \tag{34}$$

where each column is repeated for each of the $M$ B-splines. Note that, because the B-splines are local functions, the resulting matrix may be relatively sparse.

Parameter estimation is as before, but Equation (23) for the mesoscale flow is replaced by,

$$\begin{bmatrix} u_k^{\text{meso}}(t_n) \\ v_k^{\text{meso}}(t_n) \end{bmatrix} \equiv \sum_{m=1}^{M} \begin{bmatrix} u_0^m B^m(t_n) \\ v_0^m B^m(t_n) \end{bmatrix} + \frac{1}{2}\begin{bmatrix} \sigma_n^m + \delta^m & \sigma_s^m - \zeta^m \\ \sigma_s^m + \zeta^m & \delta^m - \sigma_n^m \end{bmatrix}\begin{bmatrix} (x_k(t_n) - x_0)B^m(t_n) \\ (y_k(t_n) - y_0)B^m(t_n) \end{bmatrix}. \tag{35}$$

The background flow and submesoscale flow are still recovered using Equations (24) and (25), respectively.

One of the advantages of using B-splines is that the model hierarchy is simplified. Figure 9 shows the complete model hierarchy that includes the first and second-moment fitting method, unlike Figure 3 which only showed the hierarchy for the second-moment fitting method. The key simplification is that with B-splines we can drop $(u_1, v_1)$ from $X$ when going from Equation (20) to Equation (34), since time dependence is encoded in the B-spline estimates for $(u_0, v_0)$. Choosing the appropriate model from Figure 9 proceeds exactly as in Section 3.3, but with the additional caveat that one must choose the spline degree $S$ and the number of splines $M$. With the restriction that the spline degree $S < M$, a reasonable upper bound is $S = 3$, the cubic spline. The number of splines $M$ can be chosen by assuming a minimum window length (as discussed in Section 4.2), treating the centre of each window as a data point, and then applying the formula for the canonical interpolating spline in [11]. To compute this explicitly, assume a time series of length $T$, with minimum window length $W$, then this results in a total of $M = \max(\lfloor T/W \rfloor, 1)$ evenly sized windows of minimum length. Now apply Equations (7) and (8) in [11] using pseudo points at $\{t_1, t_1 + T/M(j - 1/2), t_N\}$ where $j = 2, \ldots, M - 1$. When the drifters are evenly sampled in time, this will result in $M$ splines that each have support from the same number of data points, and each data point will intersect $S + 1$ splines. As a result, there is really only one parameter to adjust: the effective window length or, alternatively, the number of splines $M$. Because setting $M = 1$ exactly reproduces the approach in Section 3.1 using fixed parameters, the freedom for parameters to vary over time can be systematically increased by increasing $M$.

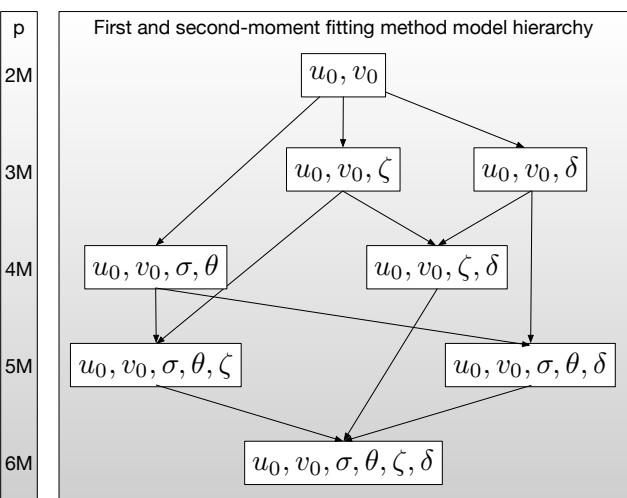

**Figure 9.** Hierarchy of first and second-moment mesoscale models where *p* indicates the number of parameters. A model with increased complexity is used only if it explains significantly more variance than the lower complexity model. Models with fewer parameters are favoured when a choice must be made.

Quantifying uncertainty with spline solutions requires a modification to the approach in Section 4.1. This is because the resulting bootstrapped parameter estimates are no longer pointwise estimates of each parameter, but rather time-varying global solutions. This means that computing the mean of each mesoscale parameter at each instant in time will not, in general, result in a valid solution since each solution is a global fit to the data. As a result, rather than considering a mean value from bootstrap solutions, as in Figure 5, we must establish the most likely bootstrap solution. Applying the bootstrap *B* times results in *B* continuous time varying model solutions of the parameters. Thus, we compute the most likely solution (of the *B* solutions) from an estimated joint probability distribution function (PDF). Specifically, for each estimated parameter in the model, we use a kernel density estimator to estimate a PDF from the bootstrap replicates for each parameter at each point in time using the methodology in [20]. For example, at time $t_n$ we estimate a one-dimensional PDF $\hat{P}_\zeta(t_n, \hat{\zeta})$ using the *B* bootstrap parameter estimates for $\zeta$ and a two-dimensional PDF $\hat{P}_{\sigma_n, \sigma_s}(t_n, \hat{\sigma}_n^b(t_n), \hat{\sigma}_s^b(t_n))$ for $\sigma_n, \sigma_s$. The likelihood of each path is then found with

$$L(\hat{\sigma}_n^b, \hat{\sigma}_s^b, \hat{\zeta}^b) = \prod_{n=1}^{N} \hat{P}_{\sigma_n, \sigma_s}(t_n, \hat{\sigma}_n^b(t_n), \hat{\sigma}_s^b(t_n)) \cdot \hat{P}_\zeta(t_n, \hat{\zeta}^b(t_n)), \tag{36}$$

where, in practice, we include probabilities from all estimated parameters. The most likely solution is that with maximum *L*, where confidence intervals are similarly calculated by including the Y percent of the *B* most likely solutions.

## 5. Application to the Latmix Experiment

The lateral mixing (LatMix) field campaign of 2011 [1,21] deployed drifters and dye with the aim of understanding what causes mixing at the submesoscale, and how this varies both spatially and temporally. The experiment consisted of two drifter deployments in the Sargasso Sea, where the drifters were deployed in a cluster. The first deployment, which we refer to as 'Site 1', consisted of nine drifters tracked for 6.1 days in an area of low strain, and the second deployment, 'Site 2', consisted of eight drifters tracked for 6.3 days in an area of moderate strain. There has been a large amount of interest and research from the experiment, e.g., [22].

In Figure 10 we plot the drifter trajectories for each site both in terms of their $\{x, y\}$ positions, but also with respect to the time-varying centre-of-mass across drifters. The effect of the mesoscale, especially strain, can already be seen visually by inspecting this plot,

both in the absolute and centre-of-mass reference frames. There are also possible signs of divergence in Site 1 (the drifters spreading in a non-random way), and vorticity in Site 2. We will now inspect this in more rigorous statistical detial using the methodology of this paper.

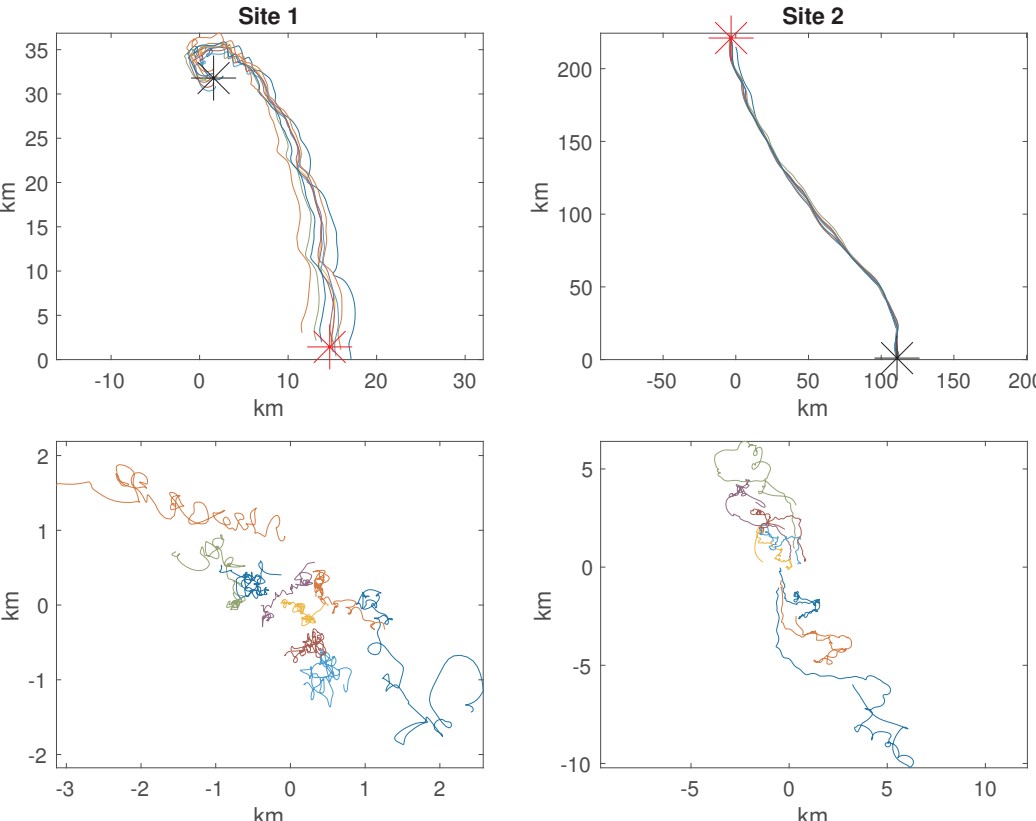

**Figure 10.** LatMix trajectories of Site 1 (nine drifters) and Site 2 (eight drifters). Top row are the positions in $\{x_k(t), y_k(t)\}$, bottom row are relative to centre-of-mass $\{\bar{x}_k(t), \bar{y}_k(t)\} = \{x_k(t) - \frac{1}{K}\sum_{k=1}^{K} x_k(t), y_k(t) - \frac{1}{K}\sum_{k=1}^{K} y_k(t)\}$. The black and red star in the top row of plots indicate the respective starting and ending centre-of-mass positions. $\{0,0\}$ in the $\{x,y\}$ components corresponds to $\{-73.0234, 31.7424\}$ degrees longitude-latitude for Site 1 and $\{-73.6776, 32.2349\}$ degrees longitude-latitude for Site 2.

### 5.1. Fixed Mesoscale Parameter Estimates

We first fit fixed (i.e., non-time-varying) mesoscale parameters to Equation (4) at each site using the second-moment fitting method described in Section 3. We present the results in the top half of Table 2 using several model hierarchies. For each model hierarchy we present the estimated mesoscale quantities, and the resulting submesoscale diffusivity. We also present FVU and FDU values (Equations (27) and (29) respectively) to assess model fit, where we remind the reader that lower values correspond to model fits with reduced error. To select the best model we use the conceptual approach illustrated earlier in Figure 3.

For Site 1 we see reasonable evidence for adding the parameters $\{\sigma, \theta\}$ ahead of vorticity $\zeta$ or divergence $\delta$, as this creates the lowest FDU values thereby creating low submesoscale diffusivities of $\kappa \approx 0.2\text{m}^2/\text{s}$, as reported in [1]. Next, we follow the hierarchy and consider adding vorticity or divergence to the strain. Here we see little evidence for vorticity, but some for divergence, with a marginal reduction in the FDU value for the latter. Finally, just for completion, we show the full hierarchy. While this full hierarchy will always yield the lowest FVU compared to all simpler models (as this is the objective function being minimised)—the FVU value does not appear to drop significantly, and the FDU value has

in fact increased, suggesting this to be an overfitted choice if we are only selecting among fixed mesoscale parameters.

For Site 2 we see mixed evidence for either initially adding divergence or strain, but the vorticity-only fit performs poorly and in fact adds diffusivity as compared to raw centre-of-mass velocities. As divergence is only one parameter (vs two for strain), we would normally proceed this way down the hierarchy using Figure 3. However, as we shall see when we account for time-variation in the mesoscale parameters, there will be more evidence for a strain-only model than a divergence-only model, therefore for comparison we follow this route down the hierarchy. When considering adding vorticity or divergence, then now there is interestingly more evidence for vorticity, with reduced FVU and FDU values. Overall however, we note that diffusivity values are much larger at Site 2 using fixed parameters, with $\kappa \approx 2\mathrm{m}^2/\mathrm{s}$. This is likely due to the presence of time-varying mesoscale features not being account for, as we shall now explore.

**Table 2.** LatMix submesoscale diffusivity estimates and associated FVU and FDU, estimated over candidate models in the hierarchy at each site using either fixed, rolling window, or spline parameter estimates. For fixed estimates we also show the mesoscale parameter estimates (scaled by the inertial frequency, $f_0$). The fixed and rolling-window estimates use the second-moment fitting method, whereas the spline estimates uses the first and second-moment fitting method.

| | | | Fixed Estimates (Site 1) | | | | |
|---|---|---|---|---|---|---|---|
| model | $\sigma$ ($f_0$) | $\theta$ (°) | $\zeta$ ($f_0$) | $\delta$ ($f_0$) | $\kappa$ (m$^2$/s) | FVU | FDU |
| $\{\zeta\}$ | 0 | 0 | −0.000137 | 0 | 0.974 | 1.000 | 1.001 |
| $\{\delta\}$ | 0 | 0 | 0 | 0.0493 | 0.361 | 0.983 | 0.371 |
| $\{\sigma,\theta\}$ | 0.0591 | −27.8 | 0 | 0 | 0.188 | 0.976 | 0.193 |
| $\{\sigma,\theta,\zeta\}$ | 0.0785 | −15.3 | −0.0443 | 0 | 0.229 | 0.971 | 0.235 |
| $\{\sigma,\theta,\delta\}$ | 0.0489 | −25.6 | 0 | 0.0137 | 0.174 | 0.976 | 0.179 |
| $\{\sigma,\theta,\zeta,\delta\}$ | 0.0711 | −12.2 | −0.0443 | 0.0137 | 0.216 | 0.971 | 0.221 |
| | | | **Fixed Estimates (Site 2)** | | | | |
| model | $\sigma$ ($f_0$) | $\theta$ (°) | $\zeta$ ($f_0$) | $\delta$ ($f_0$) | $\kappa$ (m$^2$/s) | FVU | FDU |
| $\{\zeta\}$ | 0 | 0 | 0.00613 | 0 | 4.011 | 0.999 | 1.000 |
| $\{\delta\}$ | 0 | 0 | 0 | 0.0125 | 1.886 | 0.997 | 0.470 |
| $\{\sigma,\theta\}$ | 0.0131 | −67.0 | 0 | 0 | 1.906 | 0.996 | 0.475 |
| $\{\sigma,\theta,\zeta\}$ | 0.0642 | 78.0 | 0.0650 | 0 | 1.950 | 0.985 | 0.486 |
| $\{\sigma,\theta,\delta\}$ | 0.0107 | −67.9 | 0 | 0.00258 | 1.874 | 0.996 | 0.467 |
| $\{\sigma,\theta,\zeta,\delta\}$ | 0.0637 | 77.0 | 0.0650 | 0.00258 | 1.919 | 0.985 | 0.478 |

| | Rolling Estimates (Site 1) | | | Rolling Estimates (Site 2) | | |
|---|---|---|---|---|---|---|
| model | $\kappa$ (m$^2$/s) | FVU | FDU | $\kappa$ (m$^2$/s) | FVU | FDU |
| $\{\zeta\}$ | 0.995 | 0.992 | 1.022 | 2.924 | 0.872 | 0.729 |
| $\{\delta\}$ | 0.325 | 0.974 | 0.334 | 2.341 | 0.838 | 0.584 |
| $\{\sigma,\theta\}$ | 0.183 | 0.961 | 0.188 | 1.680 | 0.710 | 0.419 |
| $\{\sigma,\theta,\zeta\}$ | 0.282 | 0.937 | 0.290 | 0.825 | 0.675 | 0.206 |
| $\{\sigma,\theta,\delta\}$ | 0.147 | 0.966 | 0.151 | 1.753 | 0.704 | 0.437 |
| $\{\sigma,\theta,\zeta,\delta\}$ | 0.248 | 0.941 | 0.255 | 0.722 | 0.669 | 0.180 |
| | **Spline Estimates (Site 1)** | | | **Spline Estimates (Site 2)** | | |
| model | $\kappa$ (m$^2$/s) | FVU | FDU | $\kappa$ (m$^2$/s) | FVU | FDU |
| $\{\zeta\}$ | 1.742 | 1.025 | 1.791 | 3.059 | 0.973 | 0.697 |
| $\{\delta\}$ | 0.342 | 0.983 | 0.352 | 3.438 | 0.831 | 0.783 |
| $\{\sigma,\theta\}$ | 0.178 | 0.976 | 0.183 | 2.118 | 0.837 | 0.483 |
| $\{\sigma,\theta,\zeta\}$ | 1.433 | 0.997 | 1.473 | 1.041 | 0.808 | 0.237 |
| $\{\sigma,\theta,\delta\}$ | 0.159 | 0.974 | 0.163 | 2.501 | 0.783 | 0.570 |
| $\{\sigma,\theta,\zeta,\delta\}$ | 1.446 | 0.996 | 1.487 | 1.466 | 0.770 | 0.334 |

*5.2. Time-Evolving Parameters Using Rolling Windows*

We now apply the rolling-window estimates using the second-moment fitting method, as discussed in Section 4.2. To pick a suitable window length $W$, we see from Table 2 that diffusivity scales as order $0.1-1$ m$^2$/s, and the strain rate when converted to days is approximately $1/3$ days. Therefore, using Figure 8 as a guide we choose a window length of $W = 1$ day (corresponding to 49 observations over 30 min sampling intervals for each drifter). This choice also coincides approximately with the inertial and diurnal periods meaning inertial oscillations and tides will be relatively close to zero mean within the window, thus being closer to satisfying the zero-mean assumption of the average submesoscale residuals across drifters made in Equations (2)–(4).

Within Table 2 we provide the estimated submesoscale diffusivity, and FVU and FDU error metrics, using rolling one-day windowed mesoscale parameter estimates for each hierarchy. As expected, the FVU decreases everywhere (as more parameters are being fitted) in comparison to the fixed-parameter fits. The FDU values, on the other hand, decrease in some but not all cases, providing mixed evidence for time-variation. We notice the reductions in FVU and FDU are most pronounced for Site 2, indicating this is the site most likely to have a time-evolving mesoscale. Overall, there is now evidence for a time-varying strain-vorticity model. Including divergence is now a less favourable choice than with the earlier analysis with fixed estimates.

In Figure 11 we display some examples of the time-varying parameter estimates using this approach. In the top panels we show the strain rate over time at each respective site using a strain-only model, where the evidence for temporal evolution at Site 2 is clear. We overlay bootstrap trajectories of these time series (as well as the fixed parameter estimates from Table 2) which indicates this variation appears significant at Site 2, but largely not at Site 1. Furthermore, the low values for strain rate of $\approx 0.01 f_0$ in the fixed-parameter estimate appears to be a misfit due to model misspecification from not allowing time-variation. The values for the strain rate are now larger at Site 2 than at Site 1 when allowing time evolution, as expected. In the bottom panels we show the time-varying strain rate and vorticity estimates using a strain-vorticity model. Again there is evidence for time-variation which we will explore further with spline fitting.

Although the parameter estimates obtained using rolling windows are overfitted and not slowly varying, these fits however provide an extremely useful lower bound, in terms of interpreting estimated submesoscale diffusivities and FVU/FDU values. This will help guide the implementation for modelling time-variation more smoothly using significantly fewer parameters in the spline methodology that follows. In contrast, the fixed parameter estimates provide a useful upper bound on diffusivities and FVU/FDU values, as this approach is the most parsimonious.

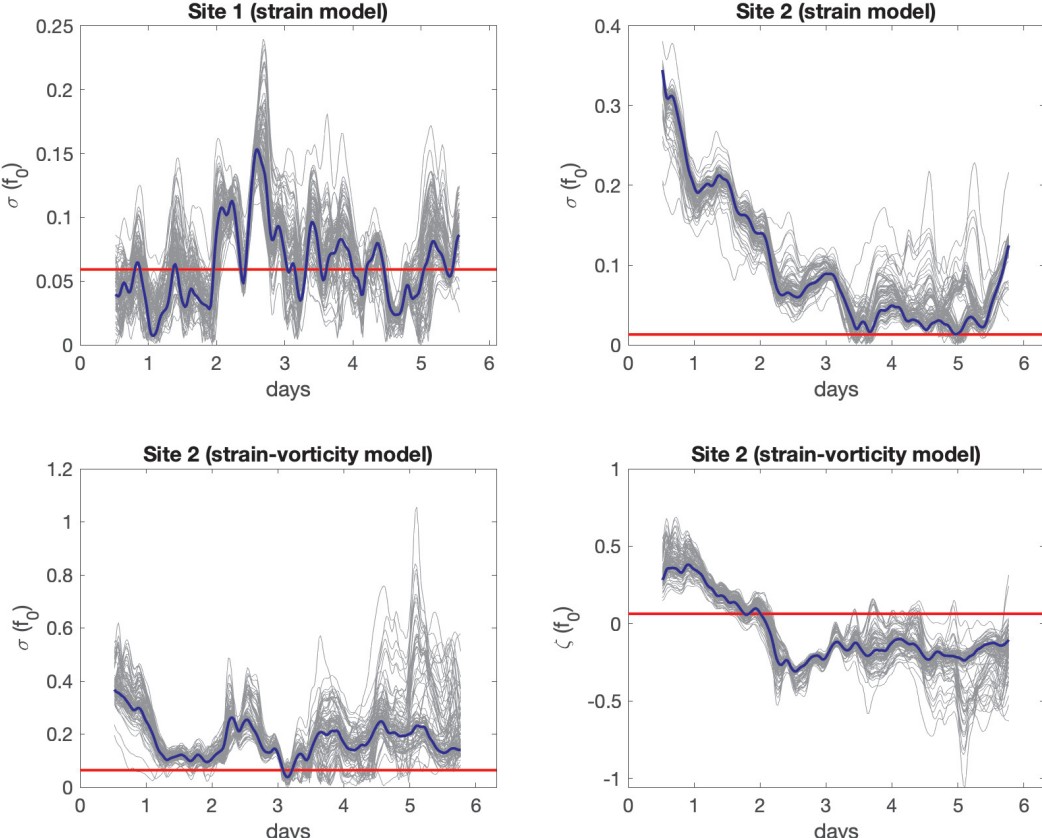

**Figure 11.** Fixed (red) and time-varying (blue) parameter estimates, where the latter are generated with a one-day rolling window using the second-moment fitting method. Top-Left: strain rate estimates with the strain-only model (Site 1). Top-Right: strain rate estimates with the strain-only model (Site 2). Bottom-Left: strain rate estimates with the strain-vorticity model (Site 2). Bottom-Right: vorticity estimates with the strain-vorticity model (Site 2). 100 bootstrapped time-varying trajectories are shown in grey in each subplot.

### 5.3. Slowly-Evolving Parameters Using Splines

We continue our analysis of the LatMix data by fitting time-evolving mesoscale parameters using the splines approach defined in Section 4.3. We will use the full first and second-moment fitting method allowing us to make a complete decomposition of the flow at both sites into background, mesoscale, and submesoscale components.

First, in Figure 12 we compare estimates of strain rate between the second-moment and the first and second-moment fitting methods during the first two days of the LatMix Site 1 experiment. This particular window has relatively low strain rates that may not be distinguishable from zero, as seen in the top-left panel of Figure 11. Using the bootstrap estimates and a kernel density estimator, the left panel of Figure 12 shows the distribution of strain rates using the second-moment fitting method. While the peak of the distribution is consistent with the strain rate estimated over the entire six day experiment, the 90% contour of the distribution includes an enormous range of strain rates, including zero. In contrast, by including the first-moment as part of the fitting method, the right-panel of Figure 12 shows a narrower range of strain rates that do not include zero. Thus, at least in this example, the combined first and second-moment fitting method provides more robust estimation than the second-moment fitting method by including extra information in the fit.

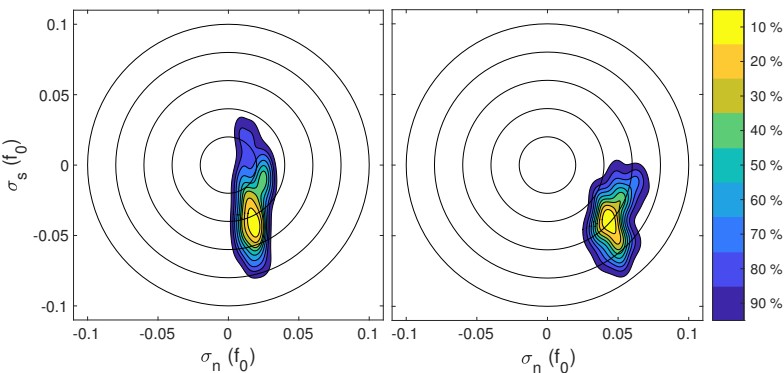

**Figure 12.** Distribution of strain rate parameters estimated for the first two days of the LatMix experiment at Site 1. Contours indicate the percentage of samples enclosed. The left panel shows estimated strain rate parameters using only the second-moment fitting method, where the right panel shows estimates using the first and second-moment fitting method.

In Figure 13 we display the time-evolving parameter estimates at Sites 1 and 2 using a strain-only and strain-vorticity model respectively. We overlay confidence intervals obtained using the bootstrap procedure outlined in Section 4.3. The time evolution of the strain-vorticity parameters is clear at Site 2, where all three mesoscale parameters $\{\sigma, \theta, \zeta\}$ are seen to change in a smooth fashion across the 6 days. In contrast, at Site 1, evidence of time variability for the strain rate is less clear, as the estimate of constant strain rate (dashed-line) fits entirely within the confidence intervals. Figure 13 also shows estimates of $\{u_0, v_0\}$, but their particular values are not directly interpretable, as they depend on the location of the expansion point, $\{x_0, y_0\}$. Instead, from Equation (3), it can be seen that they contribute to the mesoscale description of the flow at the location of the centre-of-mass.

We include the submesoscale diffusivity estimates, as well as FVU and FDU values, in the bottom portion of Table 2, along with comparison values from a hierarchy of models at each site. What we observe is quite remarkable: we can achieve FVU and FDU values that are very close to the rolling window estimates, despite using significantly fewer parameters to describe the evolution of the mesoscale velocity field. The evidence from Table 2 continues to support the choice of a strain model at Site 1 (with minor evidence for the additional presence of divergence), and a strain-vorticity model at Site 2. The estimated submesoscale diffusivities after performing the fits are around $\kappa = 0.2$ m$^2$/s at Site 1 and $\kappa = 1.0$ m$^2$/s at Site 2, nearly an order-of-magnitude difference.

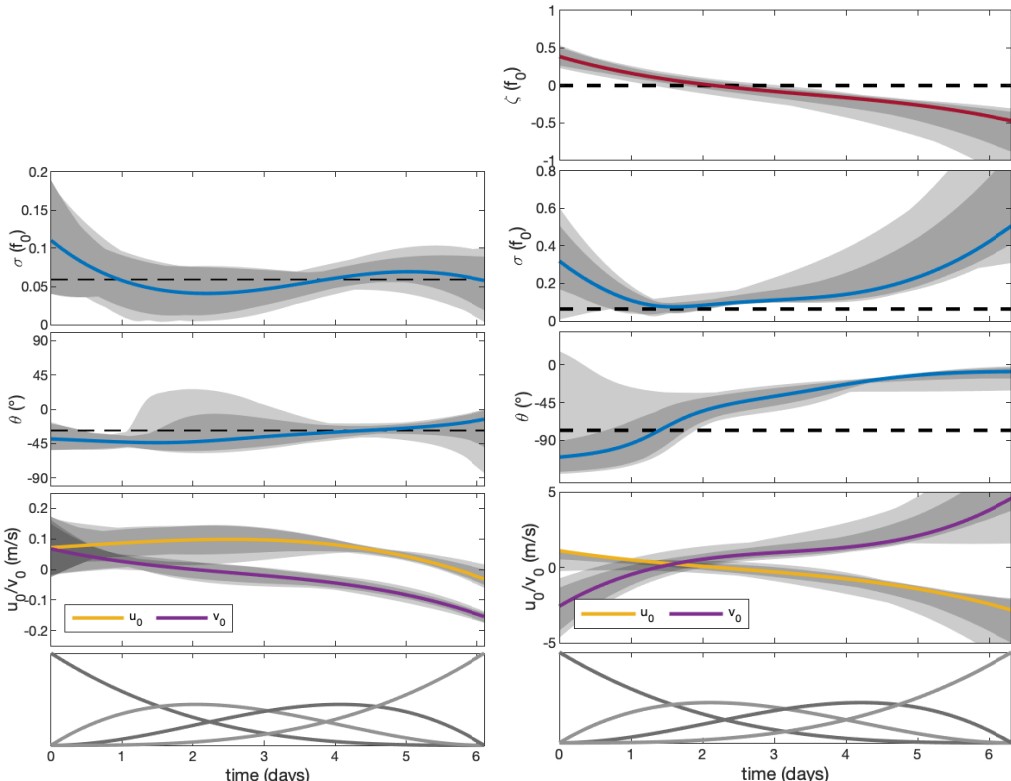

**Figure 13.** Parameters of the spline based strain model fits to Site 1 (left panel) and strain-vorticity model fits to Site 2 (right panel) using the first and second-moment fitting method. The most likely solution is highlighted, with 90% and 68% most likely solutions shown in grey and dark grey, respectively. The models are fit using four degrees of freedom per parameter with the splines shown in the bottom row.

Finally, we complete our analysis of the LatMix data by using the spline fits of Figure 13 to decompose the flow into the three components of our conceptual model of Equation (1)—background, mesoscale, and submesoscale—and then integrate over time to construct an implied set of drifter trajectories for each component. This is displayed in Figures 14 and 15 for Site 1 and Site 2 respectively. We have also included the mesoscale component in centre-of-mass coordinates. We observe that the mesoscale components meander in the fixed reference frame and follow the observed particle paths explaining most of their displacement and explain some of the spreading in the centre-of-mass frame. This can be seen by directly comparing Figures 14 and 15 with Figure 10. The submesoscale components are random-walk like and broadly resemble a diffusive process. The background components contain inertial oscillations and tides which create looping trajectories with roughly daily periodicity.

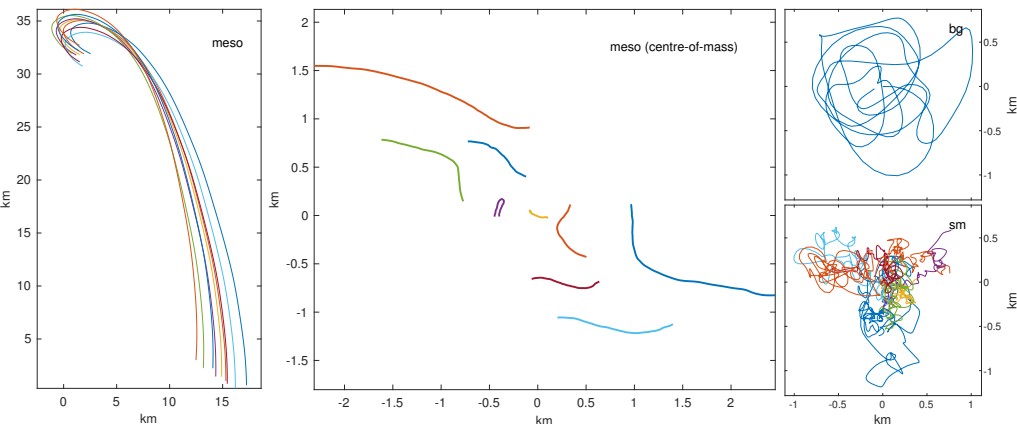

**Figure 14.** Decomposition of the flow at LatMix Site 1 using the strain-only model fitted with splines using the first and second-moment fitting method. The left panel shows the the mesoscale solution in the fixed coordinate reference frame (compare to the upper-left panel of Figure 10). The centre panel shows the same solution in the centre-of-mass frame (compare to the lower-left panel of Figure 10). The top-right and bottom-right panels show the path-integrated background and submesoscale flow, respectively.

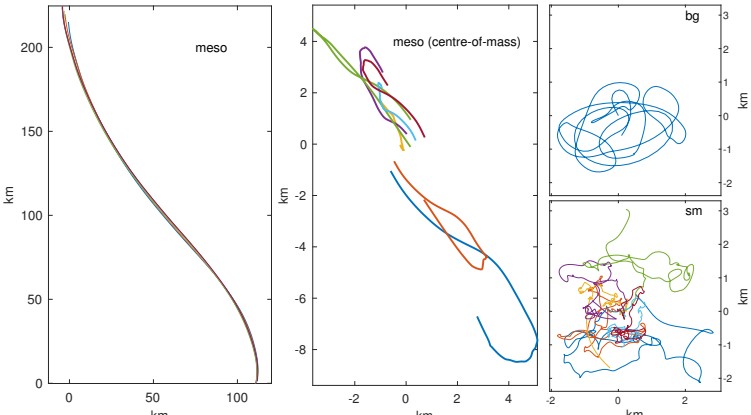

**Figure 15.** Same as Figure 14, but for LatMix Site 2 using the strain-vorticity model. The mesoscale solution in fixed frame can be compared to the upper-right panel of Figure 10), and the mesoscale solution in centre-of-mass frame can be compared to the lower-right panel of Figure 10.

Figure 16 shows the Lagrangian spectra of the background flow, the mean (across drifters) of the mesoscale flow, and the mean (across drifters) of the submesoscale flow, for Sites 1 and 2 respectively. A number of features standout in Figure 16. The Coriolis frequency is almost exactly the diurnal frequency at this latitude, and this has the effect of creating a relatively substantial peak of energy on the anticyclonic side of the spectrum of the background flow at Site 1, with no corresponding peak on the cyclonic side. This means that the oscillation is anticyclonic and nearly circular. Furthermore, the semi-diurnal tide appears primarily on the cyclonic side, although with some energy on the anticyclonic side. The background flow at Site 2 shows significantly more power, especially at lower frequencies and also has a strong inertial signal. The mesoscale flow at Site 2 is much stronger than Site 1, as expected.

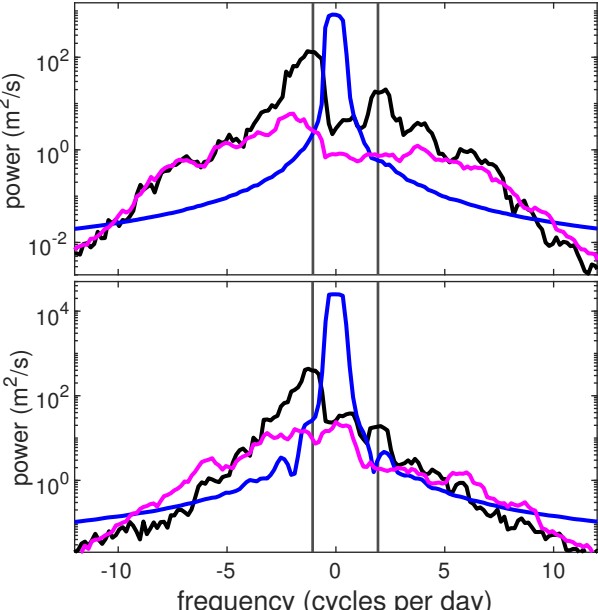

**Figure 16.** The top and bottom panels show the power spectra of the decomposed flow for Sites 1 and 2, respectively. The spectra shown are the spatially homogeneous background flow $\mathbf{u}^{bg}$ (black), the average of the mesoscale component of the flow $\mathbf{u}^{meso}$ (blue), and the average of the submesoscale component $\mathbf{u}^{sm}$ (magenta). Anticyclonic oscillations are indicated by negative frequencies and cyclonic oscillations by positive frequencies. The vertical lines indicate the semi-diurnal tidal frequency and the inertial frequency on the positive and negative side, respectively.

If the drifters were governed by the stochastic model given with Equation (9), then removing the effects of the strain in centre-of-mass coordinates would reveal a submesoscale signal given by increments of the Wiener process. The Lagrangian power spectrum would show a (flat) white noise process. However, Figure 16 shows that the submesoscale spectra from both Site 1 and 2 have significantly more structure. The spectra are characterised by low power at sub-inertial frequencies, roughly an order of magnitude more power on the anticyclonic side than the cyclonic side at near inertial frequencies, and a decay of power at higher frequencies. In our subsequent paper we will argue that these spectra are consistent with the spectrum that one would expect from internal waves.

## 6. Discussion and Conclusions

The separation in Equation (1) is a compelling conceptual model, based on the ideas of non-local spreading in turbulence theory—but is the separation actually doing something useful in practice? This idea can be tested by considering the cross-terms in the total energy of the model, as was done in [23]. Specifically, the cross terms in the kinetic energy equation,

$$\mathbf{u}_{total}^2 = \mathbf{u}_{bg}^2 + \mathbf{u}_{meso}^2 + \mathbf{u}_{sm}^2 + 2\left(\mathbf{u}_{bg}\mathbf{u}_{meso} + \mathbf{u}_{meso}\mathbf{u}_{sm} + \mathbf{u}_{bg}\mathbf{u}_{sm}\right), \tag{37}$$

should remain small if this is truly an orthogonal linear decomposition. To assess this quantity we compute the coherence between the complex submesoscale signal and the complex mesoscale signal in the centre-of-mass frame, as shown in Figure 17. The results show remarkably low coherence ($O(0.1)$) at Site 1, across all frequencies, suggesting no relation between the two signals. In contrast, Site 2 does show more coherence between the two signals, likely reflecting the challenges of the separation in time-varying conditions. Despite this, the average coherence across frequency bands is ≈0.2, suggesting the decomposition is successfully separating two mostly distinct signals. The validity of this separation can be made precise using the methodology that unambiguously separates waves and geostrophic motions at each instant in time in an Eulerian reference frame [24].

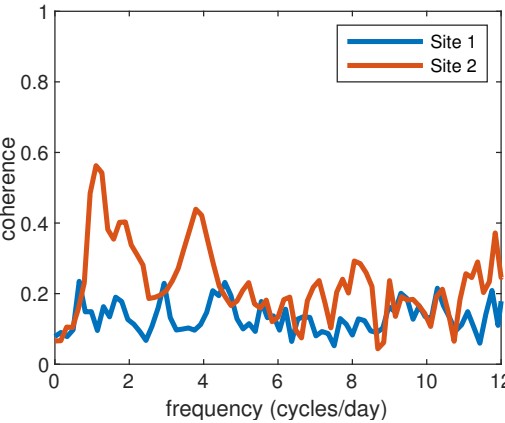

**Figure 17.** Coherence between the mesoscale signal in the centre-of-mass frame and the submesoscale signal at Site 1 and Site 2, using the disentangled velocities corresponding to the trajectories shown in Figures 14 and 15 respectively.

One of the key strengths of this methodology is how few parameters are needed to estimate the mesoscale parameters and perform the decomposition. For example, at Site 1 there are $N = 294$ observations of position from $K = 9$ drifters, resulting in $2NK$ degrees-of-freedom. The second-moment fitting method uses $2N$ degrees-of-freedom to remove the centre-of-mass. Using a single window across the entire time series to estimate the two parameters in the strain model, such as Site 1 which is well described by a single set of strain rate parameters across the entire window, leaves $2N(K-1) - 2$ degrees-of-freedom to describe the submesoscale flow. In contrast, daily rolling windows with $N_W = 49$ points (corresponding to one day) that estimate strain rate parameters at each of the $N - N_W$ time points, leaves only $2N(K-2) + 2N_W$ degrees-of-freedom to describe the submesoscale flow. As is evident in Figure 11, these extra degrees of freedom capture time-variability in the parameters that may not be appropriate. Finally, the spline fits require estimating $M$ coefficients per mesoscale parameter, and thus the spline based time-varying fits leave $2N(K-1) - 2M$ degrees-of-freedom to describe the submesoscale flow using the second-moment fits. With $M = 4$ sufficient to capture any time variability at Sites 1 and 2, this approach uses remarkably few parameters to perform this estimation. The benefit of which is that the decomposed submesoscale trajectory will contain rich statistical information with which to do further Lagrangian analysis.

As discussed in the introduction, we view this works as complementary to that of [3,10] who recently developed a method for projecting clustered drifter trajectories to local Eulerian velocity fields using Gaussian Process regression. The ultimate goal of [10] was to compute horizontal velocity gradients with which to better understand vertical transport. The method was applied to the CALYPSO an LASER drifter deployments. Applying our method to these datasets is a natural avenue for further investigation. More broadly speaking, what our method provides to complement [3,10], is not the Eulerian velocity field, but rather the Lagrangian decomposition of the trajectories into various components. This allows us to extract the specific submesoscale component from the trajectory for further analysis within the Lagrangian setting. This allows for the estimation of submesoscale diffusivity, which is not a topic covered in [3,10]. However, there is certainly scope to merge and compare our methodologies, particular because the constructed Eulerian velocity field can be directly compared with the mesoscale parameters we estimate locally over time (and hence space) using our slowly-evolving spline fits. Again, this is certainly a topic that warrants further investigation. We also see potential for our work to naturally follow-on from the recent methodology developed in [25] who identify clusters of drifter trajectories that share coherent structures. For example, such clustering could be used to divide larger deployments into smaller clusters, after which our method can then be applied to each cluster to separate flow components within coherent structures.

**Author Contributions:** Conceptualization, S.O., A.M.S. and J.J.E.; methodology, S.O., A.M.S. and J.J.E.; software, S.O., A.M.S. and J.J.E.; validation, S.O., A.M.S. and J.J.E.; formal analysis, S.O., A.M.S. and J.J.E.; investigation, S.O., A.M.S. and J.J.E.; resources, S.O., A.M.S. and J.J.E.; data curation, J.J.E.; writing—original draft preparation, S.O.; writing—review and editing, A.M.S. and J.J.E.; visualization, S.O., A.M.S. and J.J.E.; supervision, A.M.S. and J.J.E.; project administration, A.M.S. and J.J.E.; funding acquisition, A.M.S. and J.J.E. All authors have read and agreed to the published version of the manuscript.

**Funding:** The work of S. Oscroft was funded by the Engineering and Physical Sciences Research Council (Grant EP/L015692/1). The work of A. M. Sykulski was funded by the Engineering and Physical Sciences Research Council (Grant EP/R01860X/1). J. J. Early was funded by the National Science Foundation award OCE-1658564.

**Acknowledgments:** Thanks are given to Miles Sundermeyer, whose drifters were used in this analysis. Thanks are also given to Pascale Lelong for providing mentorship during the early stages of manuscript preparation.

**Conflicts of Interest:** The authors declare no conflict of interest.

## Appendix A. Sensitivity Analyses

In this section we include some supplementary simulation findings which investigate the sensitivity of the results with respect to the number of drifters in the cluster, as well as the cluster morphology (i.e., the spatial distribution of the initial deployment configuration). Our simulation results in the main body of the paper are using nine drifters configured to start as at Latmix site 1—and we used these results to motivate and help interpret our real data analysis of the Latmix data. In other drifter deployments however, the number of drifters and the configuration will vary, and we now investigate what impact this may have.

First we vary the number of drifters $K$. In Figure A1 we report the relative standard error of mesoscale parameters in the strain-dominated simulation of Figure 1. We have included a reference line that scales as $1/\sqrt{K}$ which is the asymptotic limit we expect to see standard errors reduce by according to the central limit theorem. For this simulation environment we see that the scaling behaviour is approximately correct for $K > 5$. We emphasise that in practice this scaling behaviour will not apply to real deployments. Here we have simulated drifters that experience independent submesoscale errors across drifters, which is an idealised scenario. In reality an increasing number of densely packed co-located drifters will experience correlated motions thus eventually limiting the amount of information that can be gained by adding more drifters to a cluster. Nevertheless, the simple rule from the observed scaling behaviour is that one must have approximately four times as many drifters to reduce the standard error by a factor of two.

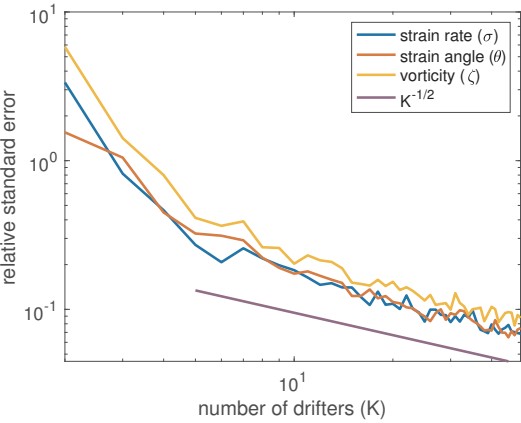

**Figure A1.** Relative standard error of stain rate, strain angle and vorticity over 100 repeated simulations for a varying number of drifters *K*. The simulation setup is as in Figure 1 in the strain-dominated model with trajectories simulated for 1 day. The initial drifter positions are sampled isotropically with expected distance to centre-of-mass fixed over all experiments to be identical to Latmix Site 1. Relative standard error is computed by dividing the observed sample standard error by the true parameter value. Therefore in this experiment we require approximately three drifters in the cluster before the standard errors are approximately half the true parameter value (and hence significantly non-zero).

We now vary the cluster morphology in our simulated environment. In Figure A2 we consider two classes of configurations. In the left panel we will contrast Latmix Site 1 configurations (in blue dots) versus two other deployment configurations: one that is parallel to the true strain angle (red dots), and another that is orthogonal to the true strain angle (green dots). To see how this affects parameter estimation we repeat the analysis of Figure 8 to find the required window length to get significant estimates of the strain rate over a range of true strain rate values—these are displayed for each configuration in the left panel of Figure A3. We see that the required window length is significantly reduced when the configuration is aligned parallel to the strain angle (red drifters), and conversely the required window length is increased when this is orthogonal (green drifters). The results with the Latmix configuration, which is more isotropic, are sandwiched in between. This analysis shows that in a strain-only field (with no vorticity or divergence), then the optimal morphology is to align drifters along the expected strain angle—but more investigation is needed to understand how the optimal configuration may change in the presence of vorticity and/or divergence, as well as background and submesoscale effects. For example, [26] showed that an isotropic configuration has the lowest error for estimating divergence, whereas configurations along a straight line, such as those in Figure A2, have the largest errors. This is in contrast to our results for a strain-only field, where the LatMix configuration is the most isotropic yet has higher error than aligning drifters along the strain angle. Therefore, the optimal morphology is dependent upon the mesoscale features present in the data, and unless these are known a priori then the best model-agnostic morphology is likely to be an isotropic cluster. We leave a thorough analysis of this for future work.

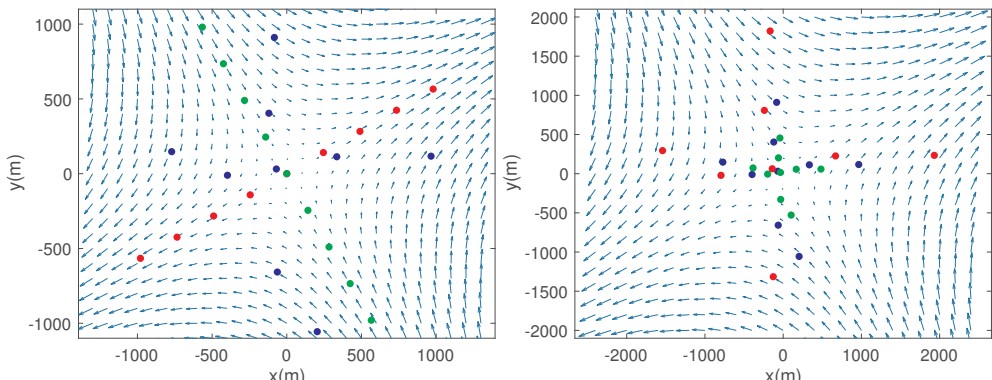

**Figure A2.** Different cluster morphologies (deployment configurations) we shall consider. In the left panel we consider nine drifters deployed as at Latmix Site 1 (blue dots), together with nine drifters deployed parallel and orthogonal to the strain angle (red and green dots respectively). In the right panel we again consider nine drifters deployed as at Latmix Site 1 (blue dots), but this time the red and green dots are the same morphologies but with the respective distances to the centre-of-mass either doubled or halved. In both panels the velocity field is as in the strain-only simulation of Figure 1, and the positions are given in centre-of-mass coordinates.

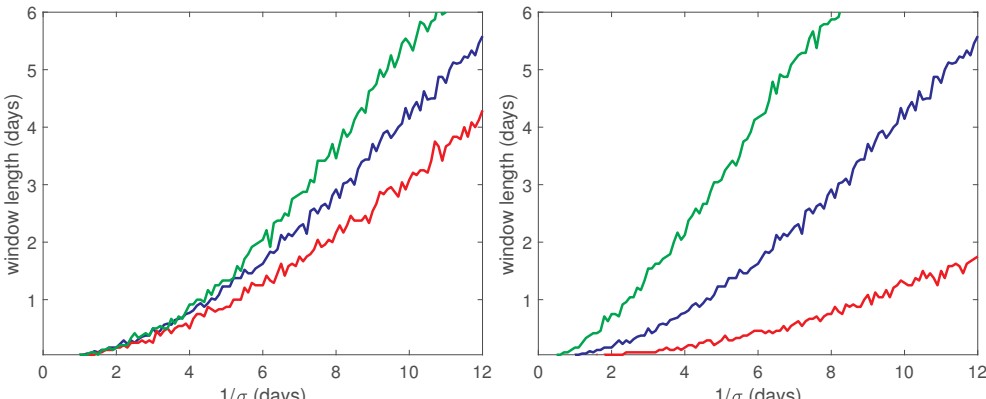

**Figure A3.** Required window lengths to obtain significant strain rate estimates for different drifter configurations. The lines in the left/right panels correspond to the drifter configurations considered in the left/right panels of Figure A2 respectively, with the colours matching the corresponding configurations. Each line corresponds to the level where the standard error of the strain rate estimate is approximately half the true strain rate value. These lines are found as in Figure 8 over 100 repeated simulations over a grid of true strain rates and window lengths.

Finally, we consider deployments where the drifters are initially configured to be closer or farther apart than in Latmix site 1, as shown in Figure A2 (right). Specifically, the red drifters are twice as far from the centre-of-mass as Latmix site 1 drifters (in blue), and the green drifters are half this distance. We repeat the same analysis over different true strain rates to find the required window lengths in the right panel of Figure A3. We observe that drifters initialised far apart require shorter window lengths to obtain significant strain rate estimates, and conversely require longer window lengths when initialised closer together. This phenomenon is easily understood in the idealised simulation scenario where spacing drifters farther provides richer information on mesoscale features as distances to centre-of-mass are increased. In practice the flow field is not homogeneous, so as with the number of drifters, there will be a practical limit as to how far apart drifters should be initially placed to ensure they are sampling the same homogeneous background flow field.

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
