# Peer review of "Separating Mesoscale and Submesoscale Flows from Clustered Drifter Trajectories"

_fluids, doi:10.3390/fluids6010014_

Round 1

Reviewer 1 Report

I believe that the presented work is a solid analytical study, proposing a variant of decomposition of the flow velocity field into sum of the background, mesoscale and submesoscale components. One of the main difficulties of such decomposition is the choice of an objective criterion for separating the components of movements of various scales. However, it makes sense to approach this issue not (exclusively) from the point of view of formal definitions, but from the point of view of practical results that can be obtained by analyzing the data of actual observations. In this sense, the approach of the authors proposing to distinguish submesoscale motions as essentially noise processes that are not explained by the model of smooth background and mesoscale phenomena has a right to exist and, as demonstrated in this work, turns out to be useful in analyzing families of clustered drifter trajectories. From a practical point of view, recommendations on the optimal number of drifters in a cluster are of particular interest.

As the authors themselves rightly point out, further analysis and comparison of the results with other works, in particular, those cited in the conclusion, are needed. However, the volume of the submitted manuscript is already quite large, and it is logical to leave the above analysis and comparison for the future.

I have no serious objection or other notes to the study. Only a careful review of the text is needed for typos, such as, for example, “our our” on line 163.

Reviewer 2 Report

A methodology is proposed for fitting a velocity field to the velocities measured by drifters when deployed in clusters. The velocity field is decomposed into background homogeneous, mesoscale and submesoscale Lagrangian components. The associated statistical uncertainty is provided. The technique is applied to drifters of the LatMix experiment. This decomposition allows to compute more meaningful estimates of submesoscale diffusivity, otherwise contaminated by the deterministic mesoscale field.

The paper is extremely well written and the figures support nicely the text. The complex mathematical methodology is clearly and fully explained. It is original and novel, and could be applied to a large amount of drifters (or floats) deployed in clusters to explore the kinematics of the ocean surface (or subsurface) currents.  I hereby recommend publication of this work in Fluids after a minor revision based on the following comments (minor revision).

General comment:

The authors should stress more the fact that the methodology can only be used for clusters of drifters with size comparable to the mesoscale (10-100 km). Furthermore, since the time-varying background field is assumed homogeneous, the drifter clusters cannot extend too much in latitude because the inertial period (and eventually the amplitude) can vary. Likewise the cluster size should be smaller than horizontal scale of tidal currents. In contrast, if the cluster is too small (<10 km) it becomes difficult to measure the mesoscale component! If in the title we change the word “Mesoscale” by “Deterministic” and “Submesoscale “ by “Stochastic” then the method is more general and there are no restrictions on the size of cluster.

More discussion is also needed on the number of drifters (K). I think that the method is also mathematically correct with K=1. Essentially we would have the mean trajectory composed on mean flow + inertial/tidal signal, mesoscale if the drifter move spatially enough to estimate the gradients which can be assumed constant say over 1 day, and the residual stochastic signal. On the other end, if K is large I guess we will have redundant dependent observations not useful for the method.

The morphology of the cluster might also be important and should be discussed. What if the drifters are and stay aligned in the cluster? The mesoscale parameters cannot be determined easily in this case. The ideal case would be an isotropic cluster, homogeneously and uniformly sampled with distance between the drifters of about 10 km in this case. Am I correct?

Another important practical parameter is the sampling period of the drifters. In this case It should be much less than 1 day. Please discuss.

Specific comments:

Caption of Fig.1. Please provide a reference for the Euler-Maruyama scheme.

Fig. 6. The mesoscale field is slowly changing so the arrows plotted in the 2 panels should be at a specific time. Which one? Initial?

Table 2. Please define fo as the inertial frequency (I suppose).

Figs 14 and 15. I would add figures showing the real Lagrangian spectra and the spectra corresponding to the reconstructed velocity components. It is a pity that you mention Lagrangian spectra in the abstract, also a lot in the methodology description, and that you do not show any real spectra with its decomposition using the method (something similar to Fig. 2 or even better Lagrangian rotary spectra!).
